# Out-Of-Distribution Detection with Diversification (Provably)

**Haiyun Yao[1], Zongbo Han[1], Huazhu Fu[2], Xi Peng[3], Qinghua Hu[1], Changqing Zhang[1]***

College of Intelligence and Computing, Tianjin University[1]
Institute of High Performance Computing, A*STAR[2]
College of Computer Science, Sichuan University[3]
{yaohaiyun, zongbo, huqinghua, zhangchangqing}@tju.edu.cn,
hzfu@ieee.org, pengx.gm@gmail.com

## Abstract

Out-of-distribution (OOD) detection is crucial for ensuring reliable deployment of machine learning models. Recent advancements focus on utilizing easily accessible auxiliary outliers (e.g., data from the web or other datasets) in training. However, we experimentally reveal that these methods still struggle to generalize their detection capabilities to unknown OOD data, due to the limited diversity of the auxiliary outliers collected. Therefore, we thoroughly examine this problem from the generalization perspective and demonstrate that a more diverse set of auxiliary outliers is essential for enhancing the detection capabilities. However, in practice, it is difficult and costly to collect sufficiently diverse auxiliary outlier data. Therefore, we propose a simple yet practical approach with a theoretical guarantee, termed Diversity-induced Mixup for OOD detection (diverseMix), which enhances the diversity of the auxiliary outlier set for training in an efficient way. Extensive experiments show that diverseMix achieves superior performance on commonly used and recent challenging large-scale benchmarks, which further confirm the importance of the diversity of auxiliary outliers. Our code is available at https://github.com/HaiyunYao/diverseMix.

## 1 Introduction

The OOD problem occurs when machine learning models encounter data that differs from the distribution of training data. In such scenarios, models may make incorrect predictions, leading to safety-critical issues in real-world applications, e.g., autonomous driving [14] and medical diagnosis [27]. To ensure the reliability of the outputs of model, it is essential not only to achieve good performance on in-distribution (ID) samples, but also to detect potential OOD samples, thus avoiding making erroneous decisions in test. Therefore, OOD detection has become a critical challenge for the secure deployment of machine learning models [1; 12; 24; 29].

Several significant studies [19; 23; 25] focus on detecting OOD examples using only ID data in training. However, due to a lack of supervision information from unknown OOD data, it is difficult for these methods to achieve satisfactory performance in detecting OOD samples. Recent methods [20; 46; 6; 34] involve training model with easily available auxiliary outliers (e.g., data from the web or other datasets), with the hope that the detection ability can generalize to unknown OOD. However, as shown in Fig. 1(a)-(b), we experimentally find that while the use of outlier datasets can enhance performance in OOD detection, the generalization capabilities of these methods remain significantly limited. Specifically, there is a considerable risk of the model overfitting to the auxiliary outliers,

---

*Corresponding authors.

38th Conference on Neural Information Processing Systems (NeurIPS 2024).

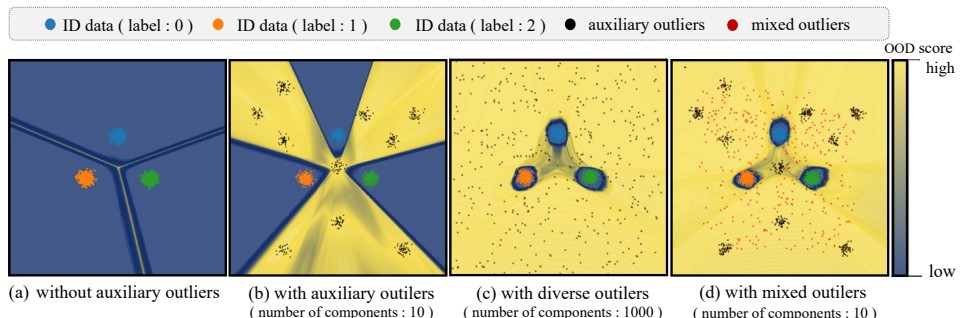

● ID data ( label : 0 )   ● ID data ( label : 1 )   ● ID data ( label : 2 )   ● auxiliary outliers   ● mixed outliers

(a) without auxiliary outliers

(b) with auxiliary outliers
( number of components : 10 )

(c) with diverse outliers
( number of components : 1000 )

(d) with mixed outliers
( number of components : 10 )

Figure 1: OOD score for different training strategies. The ID data $\mathcal{X}_{in} \subset \mathbb{R}^2$ is sampled from three distinct Gaussian distributions, each representing a different class. The auxiliary outliers are sampled from a Gaussian mixture model away from the ID data, where the number of mixture components indicates the number of classes contained in auxiliary outliers. (a) The model trained without auxiliary outliers fails to detect OOD. (b) Incorporating a less diverse set of auxiliary outliers (10 classes) during training enables partial OOD detection, but overfits auxiliary outliers. (c) OOD detection is improved with a more diverse set of auxiliary outliers (1000 classes). (d) diverseMix enriches the diversity of outliers (10 classes) through creating significantly distinct mixed outliers.

consequently failing to identify OOD samples that deviate markedly. The above limitation motivates the following important yet under-explored question: *What are the theoretical principles underlying these methods that enable better utilization of outliers?*

In this work, we theoretically investigate this crucial question from the perspective of generalization ability [3]. Specifically, we first conduct a theoretical analysis to demonstrate how the distribution shift between auxiliary outlier training set and test OOD data affects the generalization capability of OOD detector. Accordingly, a generalization bound is induced on the test-time OOD detection error of classifier, considering both empirical error and the error caused by the distribution shift between test OOD data and auxiliary outliers. Based on the theory, we deduce an intuitive conclusion that *a more diverse set of auxiliary outliers can reduce the distribution shift error and effectively lower the upper bound of the OOD detection error.* As shown in Figure 1(b)-(c), the model trained with a more diverse set of auxiliary outliers achieves better OOD detection. However, in practice, it is difficult and costly to collect sufficiently diverse outlier data. Therefore, a natural question arises - *how to guarantee the effective utilization of a **fixed set** of auxiliary outliers?*

Inspired from the theoretical principles, we propose a simple yet effective method called Diversity-induced Mixup (diverseMix) for OOD detection, which introduces and improves the mixup strategy to enhance the outlier diversity. Specifically, diverseMix employs semantic-level interpolation to generate mixed samples, creating new outliers that significantly deviate from their original counterparts. Given the risk that a random interpolation strategy (merely sampling from a predefined prior distribution) might produce mixed outliers that are unhelpful for the model (as the model can already detect them effectively), diverseMix dynamically adjusts its interpolation strategy based on original samples. This adjustment ensures that the generated outliers are novel and distinct from those previously encountered by the model, thereby enhancing diversity throughout the training process. As shown in Figure 1(b)-(d), diverseMix effectively boosts the diversity of auxiliary outliers, leading to improved OOD detection performance. The contributions of this paper are summarized as follows:

- We provide a theoretical analysis of the generalization error linked to methods trained with auxiliary outliers. By establishing an upper bound for expected error, we reveal the connection between auxiliary outlier diversity and the upper bound of OOD detection error. Our theoretical insights emphasize the importance of leveraging diverse auxiliary outliers to enhance the generalization capacity of the OOD detector.

- Constrained by the prohibitive cost of collecting outliers with sufficient diversity, we propose the Diversity-induced Mixup (diverseMix) for OOD detection, a simple yet effective strategy which is theoretically guaranteed to improve OOD detection performance.

- The proposed diverseMix achieves state-of-the-art OOD detection performance, outperforming existing methods on both standard and recent large-scale benchmarks. Remarkably, our method exhibits significant improvements over advanced methods, showing relative

performance improvements of $24.4\%$ and $43.8\%$ (in terms of FPR95) on the CIFAR-10 and CIFAR-100 datasets, respectively.

## 2    Related Works

We provide a brief review of prior research relevant to our work followed by a comparison.

**Auxiliary-Outlier-Free OOD Detection.** One early work by [19] pioneered the field of OOD detection, introducing a baseline method based on maximum softmax probability. However, it has since been established, as noted by [36], that this approach is not quite suitable for OOD detection. To address this, various methods have been developed that operate in the logit space to enhance OOD detection. These include ODIN [25], energy score [46; 28; 45], ReAct [40], logit normalization [48], Mahalanobis distance [23], and KNN-based scoring [41]. However, post-hoc OOD detection methods that do not involve pre-training on a substantial dataset generally exhibit poorer performance compared to methods that leverage auxiliary datasets for model regularization [13].

**OOD Detection with Auxiliary Outliers.** Recent advancements in OOD detection have focused on incorporating easily available auxiliary outliers into the model regularization process. Outlier exposure [20] encourage models to predict uniform distributions for outliers, and Energy-based learning [46] widens the energy gap between ID and OOD distribution. However, performance heavily depends on outlier quality. ATOM [6], POEM [34], and DOS [22] enhance performance by improving the sampling strategy for auxiliary outliers. DivOE [59] and DAL [47] improve outlier quality in a learnable manner, either in the sample space or feature space, respectively. Additionally, DOE implicitly enhances outlier informativeness through model perturbation. Incorporating outliers during training often achieves superior performance, as shown in many other works [46; 39; 2; 48].

**Comparison with Existing Methods.** Several existing methods have explored the utilization of mixup in OOD detection. MixOE [56] and OpenMix [58] perform mixup between ID data and outliers, linearly representing the transition from ID to OOD and thus enhancing the model capturing the uncertainty from outliers. Meanwhile, MixOOD [51] employ mixup on ID data to generate outliers for training. Different from existing research which primarily focuses on refining mixup strategy or designing outlier regularization method, we place emphasis on the theoretical significance of auxiliary outlier diversity. Our approach advances this concept by enhancing outlier diversity via mixup based strategy, guaranteed by a robust theoretical framework. This focus on enhancing the diversity of auxiliary outliers distinguishes our research from prevailing studies in this area.

## 3    Theory: Diverse Auxiliary Outliers Boost OOD Detection

In this section, we lay the foundation for our analysis of OOD detection. We begin by introducing the key notations for OOD detection in Sec. 3.1. In Sec. 3.2, we establish a generalization bound which highlights the critical role for auxiliary outliers in influencing the generalization capacity of OOD detection methods. Finally, in Sec. 3.3, we demonstrate how a diverse set of auxiliary outliers effectively mitigate the distribution shift errors, consequently lowering the upper bound of error. For detailed proofs, please refer to *Appendix A*.

### 3.1    Preliminaries

We consider the multi-class classification task and each sample in the training set $\mathcal{D}_{id} = \{(x_i, y_i)\}_{i=1}^{N}$ is drawn i.i.d. from the joint distribution $\mathcal{P}_{\mathcal{X}_{id} \times \mathcal{Y}_{id}}$, where $\mathcal{X}_{id}$ denotes the input space of ID data, and $\mathcal{Y}_{id} = \{1, 2, \ldots, K\}$ represents the label space. OOD detection can be formulated as a binary classification problem to learn a hypothesis $h$ from hypothesis space $\mathcal{H} \subset \{h : \mathcal{X} \to \{0, 1\}\}$ such that $h$ outputs 1 for any $x \in \mathcal{X}_{id}$ and 0 for any $x \in \mathcal{X}_{ood}$, where $\mathcal{X}_{ood} = \mathcal{X} \setminus \mathcal{X}_{id}$ represent the input space of OOD data and $\mathcal{X}$ represents the entire input space in the open-world setting. To address the challenge posed by the unknown and arbitrariness of OOD distribution $\mathcal{P}_{\mathcal{X}_{ood}}$, we leverage an auxiliary dataset $\mathcal{D}_{aux}$ drawn from the distribution $\mathcal{P}_{\mathcal{X}_{aux}}$ to serve as partial OOD data, where $\mathcal{X}_{aux} \subset \mathcal{X}_{ood}$. Due to the diversity of real-world OOD data, auxiliary outliers cannot fully represent all OOD data, so $\mathcal{P}_{\mathcal{X}_{aux}} \neq \mathcal{P}_{\mathcal{X}_{ood}}$. We aim to train a model on data sampled from $\mathcal{P}_{\widetilde{\mathcal{X}}} = k_{train} \mathcal{P}_{\mathcal{X}_{id}} + (1 - k_{train}) \mathcal{P}_{\mathcal{X}_{aux}}$ to obtain a reliable hypothesis $h$ that can effectively generalize to the unknown test-time distribution $\mathcal{P}_{\mathcal{X}} = k_{test} \mathcal{P}_{\mathcal{X}_{id}} + (1 - k_{test}) \mathcal{P}_{\mathcal{X}_{ood}}$, where $k_{train}$ and $k_{test}$

determine the proportion of ID and OOD data used for training and testing, respectively. Note that $k_{test}$ is unknown due to unpredictable test data distribution.

## 3.2 Generalization Error Bound in OOD Detection

**Basic Setting.** We define an OOD label function which provides ground truth labels (OOD or ID) for inputs as $f : \mathcal{X} \rightarrow [0, 1]$. The expectation that a hypothesis $h$ disagrees with $f$ with respect to a distribution $\mathcal{P}$ is defined as:

$$\epsilon_{\mathcal{P}}(h, f) = E_{x \sim \mathcal{P}}[|h(x) - f(x)|]. \tag{1}$$

The set of ideal hypotheses on the training data distribution $P_{\widetilde{\mathcal{X}}}$ and test-time data distribution $P_{\mathcal{X}}$ is defined as:

$$\mathcal{H}^*_{aux} : h = \arg\min_{h \in \mathcal{H}} \epsilon_{P_{\widetilde{\mathcal{X}}}}(h, f), \ \mathcal{H}^*_{ood} : h = \arg\min_{h \in \mathcal{H}} \epsilon_{P_{\mathcal{X}}}(h, f), \tag{2}$$

and we define $h^*_{ood}$ and $h^*_{aux}$ as the element in $\mathcal{H}^*_{ood}$ and $\mathcal{H}^*_{aux}$, respectively, which can be denoted as $h^*_{ood} \in \mathcal{H}^*_{ood}$, $h^*_{aux} \in \mathcal{H}^*_{aux}$. Considering that $\mathcal{X}_{aux} \subset \mathcal{X}_{ood}$, it follows that $\mathcal{H}^*_{ood} \subseteq \mathcal{H}^*_{aux}$ [2], reflecting the reality that hypotheses perform well on real-world OOD data also perform well on auxiliary outliers, conditioning on that auxiliary outliers are a subset of real-world OOD data. The generalization error of an OOD detector $h$ is defined as:

$$\text{GError}(h) = \epsilon_{x \sim \mathcal{P}_{\mathcal{X}}}(h, f). \tag{3}$$

Now, we present our first main result regarding OOD detection (training with auxiliary outliers).

**Theorem 1** *(Generalization Bound of OOD Detector). We let $\mathcal{D}_{train} = \mathcal{D}_{id} \cup \mathcal{D}_{aux}$, consisting of $M$ samples. For any hypothesis $h \in \mathcal{H}$ and $0 < \delta < 1$, with a probability of at least $1 - \delta$, the following inequality holds:*

$$GError(h) \leq \underbrace{\hat{\epsilon}_{x \sim \mathcal{P}_{\widetilde{\mathcal{X}}}}(h, f)}_{empirical\ error} + \underbrace{\epsilon(h, h^*_{aux})}_{reducible\ error} + \underbrace{\sup_{h \in \mathcal{H}^*_{aux}} \epsilon_{x \sim \mathcal{P}_{\mathcal{X}}}(h, h^*_{ood})}_{distribution\ shift\ error} + \underbrace{\mathcal{R}_m(\mathcal{H})}_{complexity} + \sqrt{\frac{\ln(\frac{1}{\delta})}{2M}} + \beta, \tag{4}$$

where $\hat{\epsilon}_{x \sim \mathcal{P}_{\widetilde{\mathcal{X}}}}(h, f)$ is the empirical error. We define $\epsilon(h, h^*_{aux}) = \int |\phi_{\mathcal{X}}(x) - \phi_{\widetilde{\mathcal{X}}}(x)||h(x) - h^*_{aux}(x)|dx$ as the reducible error, where $\phi_{\mathcal{X}}$ and $\phi_{\widetilde{\mathcal{X}}}$ is the density function of $\mathcal{P}_{\mathcal{X}}$ and $\mathcal{P}_{\widetilde{\mathcal{X}}}$ respectively. $\sup_{h \in \mathcal{H}^*_{aux}} \epsilon_{x \sim \mathcal{P}_{\mathcal{X}}}(h, h^*_{ood})$ is the distribution shift error, $\mathcal{R}_m(\mathcal{H})$ represents the Rademacher complexity, and $\beta$ is some constant condition on the error related to ideal hypotheses.

Minimizing empirical risk optimizes the model $h$ to $h \in \mathcal{H}^*_{aux}$, leading to a reduction in the reducible error, which tends to zero. However, the inherent distribution shift error between auxiliary outliers and real-world OOD data remains constant and non-negligible. This limitation fundamentally restricts the generalization of OOD detection methods trained with auxiliary outliers. To address this limitation, we investigate the effect of outlier diversity on mitigating the distribution shift error.

## 3.3 Generalization with Auxiliary OOD Diversification

In this paper, the diversity refers to semantic diversity, where a formal definition is given as follows.

**Definition 1** *(Diversity of Outliers). We assume $\mathcal{X}_{aux}$ can be divided into distinct semantic groups: $\mathcal{X}_{aux} = \mathcal{X}^{y_1} \cup \mathcal{X}^{y_2} \cup \ldots \cup \mathcal{X}^{y_m}$, where each group $\mathcal{X}^{y_i}$ contains data points with label $y_i$. Considering a dataset $\mathcal{D}_{div}$ sampled from the distribution $\mathcal{P}_{\mathcal{X}_{div}}$, where $\mathcal{X}_{div} \subset \mathcal{X}_{ood}$ encompasses $\mathcal{X}_{aux}$ and an additional group $\mathcal{X}_{new} = \mathcal{X}^{y_{m+1}} \ldots \cup \mathcal{X}^{y_n}$ with different semantic compared to $\mathcal{X}_{aux}$, i.e., $\mathcal{X}_{div} = \mathcal{X}_{aux} \cup \mathcal{X}_{new}$, we define $\mathcal{D}_{div}$ is more diverse than $\mathcal{D}_{aux}$ in terms of the range of semantic classes covered.*

---

[2]We consider the hypothesis set $\mathcal{H}$ to consist of fully-connected ReLU network with width $d_m \leq n + 4$, where $n$ is the input dimension.

Suppose we could use this diverse auxiliary outliers dataset for training, the ideal hypotheses achieved by training with $\mathcal{D}_{div}$ are denoted as:

$$\mathcal{H}_{div}^* : h = \arg\min_{h \in \mathcal{H}} \epsilon_{x \sim \mathcal{P}_{\widetilde{\mathcal{X}}_{div}}}(h, f), \tag{5}$$

with $\mathcal{P}_{\widetilde{\mathcal{X}}_{div}} = k_{train}\mathcal{P}_{\mathcal{X}_{id}} + (1 - k_{train})\mathcal{P}_{\mathcal{X}_{div}}$. Because $\mathcal{X}_{aux} \subset \mathcal{X}_{div}$ holds, the hypotheses performing well on $\mathcal{P}_{\mathcal{X}_{div}}$ also perform well on $\mathcal{P}_{\mathcal{X}_{aux}}$, giving rise to $\mathcal{H}_{div}^* \subset \mathcal{H}_{aux}^*$. Consequently, we have:

$$\sup_{h \in \mathcal{H}_{div}^*} \epsilon_{x \sim \mathcal{P}_{\mathcal{X}}}(h, h_{ood}^*) \leq \sup_{h \in \mathcal{H}_{aux}^*} \epsilon_{x \sim \mathcal{P}_{\mathcal{X}}}(h, h_{ood}^*), \tag{6}$$

which indicates that training with a more diverse set of auxiliary outliers can reduce the distribution shift error. Furthermore, effective training leads to sufficient small empirical error and reducible error, and the intrinsic complexity of the model remains constant. Consequently, a more diverse set of auxiliary outliers results in a lower generalization error bound. This theorem is formalized as:

**Theorem 2** *(Diverse Outliers Enhance Generalization). Let $\mathcal{O}(GError(h))$ and $\mathcal{O}(GError(h_{div}))$ represent the upper bounds of the generalization error of detector training with vanilla auxiliary outliers $\mathcal{D}_{aux}$ and diverse auxiliary outliers $\mathcal{D}_{div}$, respectively. For any hypothesis $h$ and $h_{div}$ in $\mathcal{H}$, and $0 < \delta < 1$, with a probability of at least $1 - \delta$, the following inequality holds*

$$\mathcal{O}(GError(h_{div})) \leq \mathcal{O}(GError(h)). \tag{7}$$

**Remark.** Theorem 2 highlights that the diversity of the outlier set is a critical factor in reducing the upper bound of generalization error. However, despite the fundamental improvement in model generalization achieved by increasing the diversity of auxiliary outliers, collecting a more diverse set of auxiliary outliers is expensive, and the auxiliary outliers we can use are limited in practical scenarios, which hinders the application of outlier exposure based methods for OOD detection. This raises an intuitive question: *can we enhance the diversity of a fixed outlier set for better utilization?*

## 4 Method: Diversity-induced Mixup (diverseMix)

In this section, we show how diverseMix addresses the challenge of effective training when the outlier diversity is limited. We begin with a theoretical analysis demonstrating the effectiveness of mixup in enhancing outlier diversity to improve OOD detection performance, providing a reliable guarantee for our mixup-based method. Then, we introduce a simple yet effective framework implementing our method diverseMix to enhance OOD detection performance.

### 4.1 Theoretical Insights: Semantic Interpolation Guarantees Enhanced Diversity of Outliers

Mixup [55] is a widely used machine learning technique to augment training data by creating synthetic samples, which has been extensively utilized in various studies[17; 7; 52]. It involves generating virtual training samples (referred to as mixed samples) through linear interpolations between data points and corresponding labels, given by:

$$\hat{x} = \lambda x_i + (1 - \lambda)x_j, \quad \hat{y} = \lambda y_i + (1 - \lambda)y_j, \tag{8}$$

where $(x_i, y_i)$ and $(x_j, y_j)$ are two samples drawn randomly from the empirical training distribution, and $\lambda \in [0, 1]$ is usually sampled from a Beta distribution with parameter $\alpha$ denoted as $Beta(\alpha, \alpha)$. This technique assumes a linear relationship between semantics (labels) and features (in data), allowing us to create new mixed samples that deviate significantly from the semantics of the original ones by combining features from samples with distinct semantics. These new mixed samples are situated outside of the original data manifold [16]. We summarize this assumption as follows:

**Assumption 1** *(Semantic Change under Mixup). Let $x_i$ and $x_j$ be any two data points from input spaces $\mathcal{X}^{y_i}$ and $\mathcal{X}^{y_j}$, respectively, where $y_i$ and $y_j$ are corresponding semantic labels and $y_i \neq y_j$. If $\zeta < \lambda < 1 - \zeta$, then there exists a positive value $\zeta$ such that the mixed data point $\hat{x} = \lambda x_i + (1-\lambda)x_j$ does not belong to either $\mathcal{X}^{y_i}$ or $\mathcal{X}^{y_j}$.*

This assumption suggests that we can enhance the diversity of outliers by generating new outliers with distinct semantics using mixup. Specifically, applying mixup to outliers in $\mathcal{X}_{aux}$ results in some

generated mixed outliers having different semantics, suggesting that they belong to novel (unknown or unnamed) semantic classes outside of $\mathcal{X}_{aux}$. Consequently, these mixed outliers can be considered as samples from a broader region within the input space. As per Definition 1, the mixed outliers exhibit greater diversity than the original outliers. This lemma is formally presented as follows:

**Lemma 1** *(Diversity Enhancement with Mixup). For a group of mixup transforms[3] $\mathcal{G}$ acting on the input space $\mathcal{X}_{aux}$ to generate an augmented input space $\mathcal{G}\mathcal{X}_{aux}$, defined as $\mathcal{G}\mathcal{X}_{aux} = \{\hat{x}|\hat{x} = \lambda x_1 + (1-\lambda)x_2; x_1, x_2 \in \mathcal{X}_{aux}, \lambda \in [0,1]\}$, the following relation holds:*

$$\mathcal{X}_{aux} \subset \mathcal{G}\mathcal{X}_{aux}. \tag{9}$$

Lemma 1 establishes that mixed outliers $\mathcal{D}_{mix}$ exhibits greater diversity compared to $\mathcal{D}_{aux}$, where $\mathcal{D}_{mix}$ is drawn from distribution $\mathcal{P}_{\mathcal{G}\mathcal{X}_{aux}}$. Consequently, according to Theorem 2, mixup outliers contribute to a reduction in generalization error. We can formalize this relationship as follows, and the detailed proofs can be found in *Appendix A*.

**Theorem 3** *(Mixed Outlier Enhances Generalization). Let $\mathcal{O}(GError(h))$ and $\mathcal{O}(GError(h_{mix}))$ represent the upper bounds of the generalization error of detector training with vanilla auxiliary outliers $\mathcal{D}_{aux}$ and mixed auxiliary outliers $\mathcal{D}_{mix}$, respectively. For any hypothesis $h$ and $h_{mix}$ in $\mathcal{H}$, and $0 < \delta < 1$, with a probability of at least $1 - \delta$, we have*

$$\mathcal{O}(GError(h_{mix})) \leq \mathcal{O}(GError(h)). \tag{10}$$

Theorem 3 demonstrates that mixup enhances auxiliary outlier diversity, reducing the upper bound of generalization error in OOD detection, which provides a reliable guarantee of mixup's effectiveness in improving OOD detection. However, the vanilla mixup lacks flexibility, which may generate outliers that are not necessarily beneficial to the model. Next, we will provide an implementation of our method which dynamically adjusts the interpolation strategy in a data-adaptive manner.

### 4.2 Implementation

Considering a classifier network $\theta$ and $F(x,\theta)$ denotes the logit outputs for input $x$, our goal is to use the scoring function $S(x,\theta)$ to develop an OOD detector:

$$G(x) = \text{ID} \cdot \mathbf{1}\{S(x,\theta) \geq \gamma\} + \text{OOD} \cdot \mathbf{1}\{S(x,\theta) < \gamma\}, \tag{11}$$

where $\mathbf{1}\{\cdot\}$ is the indicator function, $\gamma$ is the threshold, typically chosen to ensure that a significant proportion (e.g., 95%) of ID data is accurately identified. The training objective is given by:

$$\arg\min_\theta \mathbb{E}_{(x,y)\sim\mathcal{D}_{id}}[\mathcal{L}_{\text{CE}}(F(x,\theta), y)] + \omega \cdot \mathcal{L}_{\text{aux}}, \tag{12}$$

where $\mathcal{L}_{\text{CE}}(\cdot)$ is the cross entropy loss, $\mathcal{L}_{\text{aux}}$ serves as a regularization term enabling model to learn from auxiliary outliers with low-confidence predictions, and $\omega$ controls the strength of regularization.

Our previous analysis showed that semantic interpolation can increase the diversity of outliers, thereby enhancing the model's OOD detection performance. However, the interpolation weights in vanilla mixup is randomly sampled from a preset prior distribution (e.g. beta distribution), which may result in generating mixed outliers that are not necessarily beneficial to the model. To efficiently increase the diversity of auxiliary outliers, we dynamically adjust the mixup strategy based on the original outliers, thereby generating novel mixed outliers which are more likely to be unfamiliar to the model.

During each training epoch, outliers are regularized, prompting the model $\theta$ to assign lower scores to previously encountered outliers. Consequently, outliers that achieve higher scores $S(x,\theta)$ are more likely to be novel or previously unseen outliers. We expect the generated outliers to be located in the vicinal space of the novel outliers that have not yet been encountered by the model. To achieve this, we adjust the prior distribution based on scores. Specifically, for outlier samples $x_i$ and $x_j$ randomly drawn from the empirical auxiliary outlier distribution, the mixed outliers are formulated as follows:

$$\hat{x} = \lambda x_i + (1-\lambda)x_j, \ \lambda \sim \text{Beta}(\hat{s}_i\alpha, \hat{s}_j\alpha), \tag{13}$$

where $\hat{s}_i, \hat{s}_j$ adjusts the original Beta distribution according to $x_i$ and $x_j$, which is defined as follows:

$$\hat{s}_i = \frac{\exp(S(x_i,\theta)/T)}{\sum_{k\in\{i,j\}} \exp(S(x_k,\theta)/T)}, \tag{14}$$

---

[3]The set of all possible combinations of data points and all possible values of $\lambda$ for mixup.

**Algorithm 1:** diverseMix for OOD Detection

---

**Input:** ID dataset $\mathcal{D}_{\mathrm{id}}$, outlier dataset $\mathcal{D}_{\mathrm{aux}}$, batch size $N$, distribution parameter $\alpha$, temperature $T$.

**Output:** model parameters $\theta$.

**for** *each iteration* **do**
   **for** *each mini-batch* **do**
      Sample $N$ ID data from $\mathcal{D}_{\mathrm{id}}$ as $\mathcal{B}_{\mathrm{id}}$ and $N$ outliers from $\mathcal{D}_{\mathrm{aux}}$ as $\mathcal{B}_{\mathrm{aux}}$, respectively.
      Evaluate the auxiliary outliers $\mathcal{B}_{\mathrm{aux}}$ using the current model $\theta$ to obtain the scores $\mathcal{S}$.
      Randomly shuffle $\mathcal{B}_{\mathrm{aux}}$ and the corresponding scores $\mathcal{S}$ to generate $\mathcal{B}'_{\mathrm{aux}}$ and $\mathcal{S}'$.
      Generate prior adjustment strategies based on scores $\mathcal{S}$ and $\mathcal{S}'$ according to Eq. 14.
      Sample the interpolation weight from the adjusted prior distribution and generate mixed
        outliers $\mathcal{D}_{mix}$ according to Eq. 13.
      Train the model $\theta$ using the objective function defined in Eq. 12.
   **end for**
**end for**

---

with $T$ representing the temperature parameter. This adaptive strategy assigns higher weights to the outliers that contain more information unknown to the current model, ensuring the generation of novel outliers, thereby increasing diversity throughout the training process. After constructing the mixed auxiliary outliers, they are used for the training objective (12). The whole pseudo code of the proposed method is shown in Alg. 1.

**Compatibility with different OOD regularization method.** DiverseMix is a general method that is suitable for a series of OOD regularization methods. One representative method is the energy-based method [46], which employs the following OOD regularization loss:

$$\mathcal{L}_{aux} = \mathbb{E}_{(x,y)\sim\mathcal{D}_{id}}[(\max(0, m_{in} - S(x;\theta))^2] + \mathbb{E}_{x\sim\mathcal{D}_{aux}}[(\max(0, S(x;\theta) - m_{out}))^2], \quad (15)$$

where $m_{in}$ and $m_{out}$ are margin hyperparameters, and $S(x;\theta) = \log\sum_{i=1}^{K}\exp(F_i(x,\theta))$ is the corresponding scoring function. More details for regularization methods are provided in *Appendix* B.3.

## 5 Experiments

In this section, we outline our experimental setup and conduct experiments on common OOD detection benchmarks to answer the following questions: **Q1.** Effectiveness (I): Does our method outperform its counterparts in OOD detection? **Q2.** Effectiveness (II): Does our method retain its superior performance across various settings including large-scale benchmarks? **Q3.** Practicability (I): Does our method demonstrate effectiveness across different OOD regularization methods? **Q4.** Practicability (II): Does our method demonstrate effectiveness in low-quality auxiliary outlier datasets? **Q5.** Ablation study: (I) Does diverseMix truly offer a distinct advantage over other data augmentation methods? (II) What is the key factor contributing to performance improvement in our method? **Q6.** Reliability: Do the experimental results provide strong support for established theory?

### 5.1 Experimental Setup

We briefly present the experimental setup here, including the experimental datasets and evaluation metrics. Further experimental details can be found in *Appendix B*.

**Datasets.** ○ **ID datasets.** Following the commonly used benchmark in OOD detection literature, we use *CIFAR-10*, *CIFAR-100* and *ImageNet-200* as ID datasets. ○ **Auxiliary outlier datasets.** For CIFAR experiments, the downsampled version of ImageNet (*ImageNet-RC*) is employed as auxiliary outliers. For ImageNet-200 experiments, the remaining 800 categories from ImageNet-1k (*ImageNet-800*) serve as auxiliary outliers. ○ **OOD test sets.** For CIFAR benchmark, we use diverse datasets including *SVHN* [37], *Textures* [8], *Places365* [57], *LSUN-crop*, *LSUN-resize* [53], and *iSUN* [49]. For ImageNet benchmark, We use datasets such as *SSB-hard* [43], *NINCO* [5], *iNaturalist* [42], *Textures* [8] and *OpenImage-O* [44].

**Evaluation metrics.** Following common practice, we report: (1) OOD false positive rate at 95% true positive rate for ID samples (*FPR95*) [26], (2) the area under the receiver operating characteristic

Table 1: **Main results.** Comparison with competitive OOD detection methods trained with the same DenseNet backbone. The performance metrics are averaged (%) over six OOD test datasets from Section 5.1. The best results are in **bold**. *diverseMix not only demonstrates state-of-the-art OOD detection performance on the CIFAR benchmark but also maintains high accuracy in ID classification.* More details are provided in the *Appendix B*.

| Method | CIFAR-10 | | | | CIFAR-100 | | | | w./w.o. $\mathcal{D}_{aux}$ |
| | FPR95 (↓) | AUROC (↑) | AUPR (↑) | ID-ACC | FPR95 (↓) | AUROC (↑) | AUPR (↑) | ID-ACC | |
|---|---|---|---|---|---|---|---|---|---|
| MSP | 58.98 | 90.63 | 93.18 | 94.39 | 80.30 | 73.13 | 76.97 | 74.05 | × |
| ODIN | 26.55 | 94.25 | 95.34 | 94.39 | 56.31 | 84.89 | 85.88 | 74.05 | × |
| Mahalanobis | 29.47 | 89.96 | 89.70 | 94.39 | 47.89 | 85.71 | 87.15 | 74.05 | × |
| Energy | 28.53 | 94.39 | 95.56 | 94.39 | 65.87 | 81.50 | 84.07 | 74.05 | × |
| SSD+ | 7.22 | 98.48 | 98.59 | NA | 38.32 | 88.91 | 89.77 | NA | × |
| OE | 9.66 | 98.34 | 98.55 | 94.12 | 19.54 | 94.93 | 95.26 | 74.25 | ✓ |
| SOFL | 5.41 | 98.98 | 99.10 | 93.68 | 19.32 | 96.32 | 96.99 | 73.93 | ✓ |
| CCU | 8.78 | 98.41 | 98.69 | 93.97 | 19.27 | 95.02 | 95.41 | 74.49 | ✓ |
| Energy (w. $\mathcal{D}_{aux}$) | 4.62 | 98.93 | 99.12 | 92.92 | 19.25 | 96.68 | 97.44 | 72.39 | ✓ |
| NTOM | 4.00 | 99.09 | 98.61 | 94.26 | 18.77 | 96.69 | 96.49 | 74.52 | ✓ |
| POEM | 2.54 | 99.40 | 99.50 | 93.49 | 15.14 | 97.79 | 98.31 | 73.41 | ✓ |
| MixOE | 14.54 | 97.16 | 97.41 | 94.48 | 27.71 | 92.93 | 93.81 | 75.15 | ✓ |
| DivOE | 11.41 | 97.76 | 98.18 | 93.74 | 18.91 | 95.00 | 95.26 | 74.08 | ✓ |
| DiverseMix (ours) | **1.92** | **99.42** | **99.51** | 94.16 | **8.51** | **98.24** | **98.46** | 74.60 | ✓ |

Table 2: **Main results on large-scale ImageNet benchmark.** Comparison with competitive OOD detection methods trained with the same ResNet backbone. For better presentation, the best and second-best results are in **bold** and underline respectively. *Consistent with CIFAR experiment results, diverseMix demonstrates strong OOD detection capabilities for both near-OOD and far-OOD test sets, achieving state-of-the-art OOD detection performance.* Details are provided in the *Appendix B*.

| Method | Near-OOD | | | Far-OOD | | | Average | | | ID-ACC |
| | FPR (↓) | AUROC (↑) | AUPR (↑) | FPR (↓) | AUROC (↑) | AUPR (↑) | FPR (↓) | AUROC (↑) | AUPR (↑) | |
|---|---|---|---|---|---|---|---|---|---|---|
| MSP | 70.35 | 82.75 | 88.58 | 54.51 | 88.81 | 91.86 | 60.85 | 86.39 | 90.54 | 85.81 |
| Energy | 70.35 | 81.88 | 88.56 | 53.87 | 89.30 | 91.59 | 60.46 | 86.33 | 90.38 | 85.81 |
| Max Logits | 69.45 | 82.25 | 88.69 | 52.49 | 89.60 | 92.13 | 59.28 | 86.66 | 90.75 | 85.81 |
| ODIN | 69.06 | 82.20 | 88.75 | 50.90 | 89.90 | 92.50 | 58.16 | 86.82 | 91.00 | 85.81 |
| OE | 59.12 | **86.86** | **92.69** | 54.95 | 90.51 | 91.20 | 56.61 | 89.05 | 91.79 | 85.52 |
| Energy (w. $\mathcal{D}_{aux}$) | 60.67 | 85.95 | 91.75 | 58.07 | 89.73 | 89.67 | 59.11 | 88.22 | 90.50 | 84.94 |
| DPN | 63.39 | 84.94 | 91.46 | 61.31 | 89.85 | 90.16 | 62.14 | 87.89 | 90.68 | 85.27 |
| MixOE | 68.43 | 83.42 | 88.74 | 50.51 | 90.62 | 92.31 | 57.68 | 87.38 | 90.89 | 86.35 |
| DiverseMix (ours) | 59.81 | 86.36 | 91.76 | **48.58** | **91.35** | **92.38** | **53.07** | **89.36** | **92.13** | 85.95 |

curve (*AUROC*) [10], (3) the area under the precision-recall curve (*AUPR*) [31]. We also provide ID classification accuracy (*ID-ACC*).

## 5.2 Experimental Results and Discussion

**DiverseMix achieves superior performance on the common benchmark (Q1).** Our method outperforms existing competitive methods, establishing *state-of-the-art* performance both on *CIFAR-10* and *CIFAR-100* datasets. Table 1 provides a comprehensive comparison with methods grouped into: **(1) ID-only training:** *MSP* [19], *ODIN* [25], *Mahalanobis* [23], *Energy* [46]; **(2) Utilizing auxiliary outliers:** *OE* [20], *SOFL* [35], *CCU* [32], *Energy with outliers* [46], *NTOM* [6], *POEM* [34], *MixOE* [56], *DivOE* [59]. Methods that utilizing auxiliary outliers generally achieve significantly better empirical performance on OOD detection. This implies that leveraging auxiliary outliers is essential for enhancing OOD detection performance. Our method diverseMix significantly outperforms the top baseline, reducing the FPR95 by 0.62% and 6.63% on *CIFAR-10* and *CIFAR-100*, respectively. These reductions correspond to relative error reductions of 24.4% and 43.8%. These notable improvements can be attributed to the enhanced diversity in auxiliary outliers offered by diverseMix, which lowers the generalization error bound and significantly improves the OOD detection performance.

**DiverseMix is effective on the large-scale benchmark (Q2).** Recent studies [50] have suggested that methods leveraging outlier data tend to underperform in more demanding, large-scale OOD detection tasks. To evaluate the effectiveness of our method, we conduct experiments on the ImageNet benchmark. Following[50], We categorize the OOD test set into two distinct groups: near-OOD and far-OOD. For each group, we report the average performance metrics. Furthermore, we also present the overall average performance on the OOD test sets. Table 2 illustrates that methods requiring outliers during training tend to excel in near OOD detection but fall short in far-OOD detection, sometimes even performing worse than methods that do not require outliers during training. Although MixOE improves far-OOD detection performance to some extent, it fails to fully leverage auxiliary outliers to enhance near-OOD detection. In contrast, our method not only maintains strong performance in near-OOD detection but also significantly improves performance in far-OOD scenarios. We speculate that the virtual auxiliary outlier data generated by diverseMix may be more

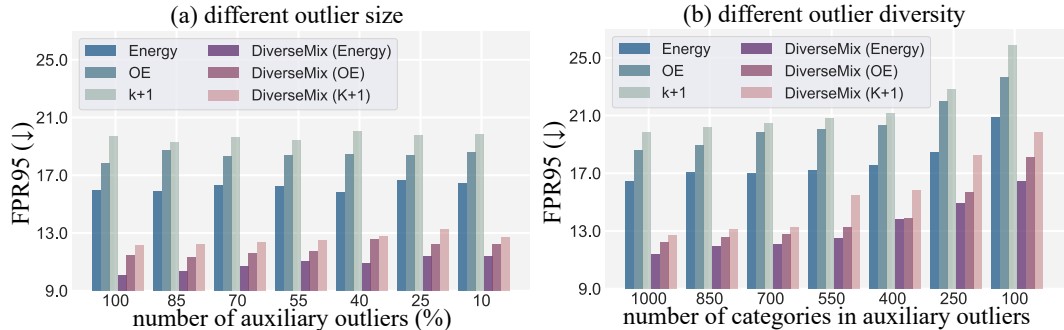

Figure 2: Comparison of OOD detection performance on CIFAR-100 with decreased quality of auxiliary outlier datasets (a) With constant diversity of auxiliary outliers (1000 categories), the dataset size is decreased. The x-axis represents the percentage of the original outlier dataset's size used for training. (b) With fixed dataset size (10% of auxiliary outliers), the diversity of outliers is decreased, with the x-axis displaying the number of categories. See *Appendix B.4* for more details.

Table 3: **Ablation study.** Performance are averaged (%) over six OOD test datasets from Section 5.1. The best results are in **bold**. More details about the comparison methods are provided in *Appendix B*

<table>
<tr><td colspan="5">(a) different data augmentation method.</td><td colspan="5">(b) different semantic interpolation strategy.</td></tr>
<tr><td></td><td colspan="2">CIFAR-10</td><td colspan="2">CIFAR-100</td><td></td><td colspan="2">CIFAR-10</td><td colspan="2">CIFAR-100</td></tr>
<tr><td></td><td>FPR (↓)</td><td>AUROC (↑)</td><td>FPR (↓)</td><td>AUROC (↑)</td><td></td><td>FPR (↓)</td><td>AUROC (↑)</td><td>FPR (↓)</td><td>AUROC (↑)</td></tr>
<tr><td>Gaussian noise</td><td>6.69</td><td>98.64</td><td>19.94</td><td>95.69</td><td>Vanilla</td><td>5.43</td><td>98.80</td><td>16.30</td><td>96.87</td></tr>
<tr><td>Cutout</td><td>7.20</td><td>98.57</td><td>19.12</td><td>96.64</td><td>Mixup</td><td>4.00</td><td>99.02</td><td>12.56</td><td>97.42</td></tr>
<tr><td>Color jitter</td><td>8.83</td><td>98.45</td><td>24.36</td><td>95.01</td><td>Cutmix</td><td>5.76</td><td>98.79</td><td>15.95</td><td>97.06</td></tr>
<tr><td>DiverseMix (ours)</td><td>**1.92**</td><td>**99.42**</td><td>**8.51**</td><td>**98.24**</td><td>DiverseMix (ours)</td><td>**1.92**</td><td>**99.42**</td><td>**8.51**</td><td>**98.24**</td></tr>
</table>

representative of far-OOD data. While most OOD detection methods face difficulties in achieving satisfactory performance across both near-OOD and far-OOD, our method excels in detecting both types of OOD, significantly surpassing other methods in the average OOD detection performance.

**DiverseMix is a general method that achieves good performance across different OOD regularization methods (Q3).** To investigate the generality of diverseMix across different OOD regularization methods, we replace the original energy loss with the *K+1* loss and the OE loss. The experimental results presented in Figure 2 reveal that diverseMix achieves consistent effectiveness regardless of the OOD regularization method employed. These findings not only suggest the versatility of our method but also provide substantial empirical evidence supporting our theoretical framework.

**DiverseMix remains effective even when the auxiliary outlier data is of low quality (Q4).** In Figure 2, the quality of auxiliary outliers used for training is decreased by gradually decreasing their quantity or their diversity. Our method diverseMix consistently outperforms previous methods by enhancing the diversity of auxiliary outliers across different dataset sizes and diversity levels. This suggests that diverseMix remains effective even when the auxiliary outliers are of low quality.

**Sample adaptive semantic interpolation contributing to unique advantages of diverseMix (Q5).** We compared diverseMix with other data augmentation methods. As shown in Table 3(a), diverseMix demonstrates superior performance for OOD detection over other data augmentation methods that preserve the semantics of outliers. Additionally, the ablation study in Table 3(b) compares diverseMix with different mixup strategies. DiverseMix outperforms both vanilla mixup and cutmix by adaptively adjusting its interpolation strategy based on the given outliers, thereby efficiently generating novel mixed outlier samples to enhance diversity. The advantages of diverseMix lie in 1) enhancing the diversity of outliers at the semantic level, and 2) efficiently boosting diversity by adaptively adjusting its strategy for the given outlier samples. For detailed comparisons, please see *Appendix B.6*.

**Our theory effectively demonstrates that the diversity of auxiliary outliers is a key factor to ensure OOD detection performance (Q6).** In Figure 2, when maintaining the diversity relatively constant and changing the quantity of data, the performance of different methods remains relatively stable. However, when the number of outliers is fixed and the diversity of the outliers dataset is reduced, there is a significant decrease in performance across all methods. This suggests that diversity is a key quality factor for the auxiliary outliers, providing substantial empirical support for our theory.

**DiverseMix has the potential for application across a wide range of task domains.** Our theory is not rely on any assumptions specific to the task domain. Given the successful implementation of

mixup across different fields [15; 54], diverseMix also has the potential for application in multiple task domains beyond just computer vision tasks. We have investigated the application of diverseMix in the NLP domain through experiments. For additional details, please see *Appendix C.2*.

## 6 Conclusions and Future Work

In this study, we demonstrate that the performance of OOD detection methods is hindered by the distribution shift between unknown test OOD data and auxiliary outliers. Through rigorous theoretical analysis, we demonstrate that enhancing the diversity of auxiliary outliers can effectively mitigate this problem. Constrained by limited access to auxiliary outliers and the high cost of data collection, we introduce diverseMix, an effective method that enhances the diversity of auxiliary outliers and significantly improves model performance. The effectiveness of diverseMix is supported by both theoretical analysis and empirical evidence. Furthermore, our theory enables future research to design new OOD detection method. We hope that our research can bring more attention to the diversity in OOD detection.

## 7 Acknowledgements

This work was supported by the National Natural Science Foundation of China (Grant No.62376193), the National Science Fund for Distinguished Young Scholars (Grant No.61925602) and the H. Fu's Agency for Science, Technology and Research (A*STAR) Central Research Fund ("Robust and Trustworthy AI system for Multi-modality Healthcare"). The authors also appreciate the suggestions from NeurIPS anonymous peer reviewers.

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

# Appendix

## Contents

## A   Theoretical Analysis

In this section, we provide detailed proofs of our theories and the proposed method, including the proof of $\mathcal{H}_{ood}^* \subseteq \mathcal{H}_{aux}^*$, the establishment of the generalization error bound for OOD detection (Theorem 1), a more diverse set of auxiliary outliers leads to a reduced generalization error (Theorem 2), and the proof of diversity enhancement with mixup (Lemma 1).

### A.1   Proof of $\mathcal{H}_{ood}^* \subseteq \mathcal{H}_{aux}^*$

In this section, we demonstrate that if $\mathcal{H}$ consist of fully-connected ReLU network with width $d_m \leq n + 4$, where $n$ is the input dimension, and given that that $\mathcal{X}_{aux} \subset \mathcal{X}_{ood}$, it follows that $\mathcal{H}_{ood}^* \subseteq \mathcal{H}_{aux}^*$, This reflects the reality that hypotheses that perform well on real-world OOD data also perform well on auxiliary outliers, conditioning on that auxiliary outliers are a subset of real-world OOD data.

**Proof.** We first express the expected error of hypotheses $h$ on the training data distribution $\mathcal{P}_{\widetilde{\mathcal{X}}}$ and the unknown test-time data distribution $\mathcal{P}_{\mathcal{X}}$ as follows:

$$
\begin{cases}
\epsilon_{\mathcal{P}_{\widetilde{\mathcal{X}}}}(h, f) = \int_{\widetilde{\mathcal{X}}} |h(x) - f(x)| dx = \int_{\mathcal{X}_{aux}} |h(x) - f(x)| dx + \int_{\mathcal{X}_{id}} |h(x) - f(x)| dx = \epsilon_1, \\
\int_{\mathcal{X}_{ood} \setminus \mathcal{X}_{aux}} |h(x) - f(x)| dx = \epsilon_2, \\
\epsilon_{\mathcal{P}_{\mathcal{X}}}(h, f) = \int_{\mathcal{X}} |h(x) - f(x)| dx = \int_{\mathcal{X}_{id}} |h(x) - f(x)| dx + \int_{\mathcal{X}_{ood}} |h(x) - f(x)| dx \\
\quad = \int_{\mathcal{X}_{id}} |h(x) - f(x)| dx + \int_{\mathcal{X}_{aux}} |h(x) - f(x)| dx + \int_{\mathcal{X}_{ood} \setminus \mathcal{X}_{aux}} |h(x) - f(x)| dx = \epsilon_1 + \epsilon_2.
\end{cases}
$$

From the above expressions, we obtain:

$$\begin{cases} \mathcal{H}_{aux}^* = \{h : \arg\min_h \epsilon_1\}, \\ \mathcal{H}_{other}^* = \{h : \arg\min_h \epsilon_2\}, \\ \mathcal{H}_{ood}^* = \{h : \arg\min_h (\epsilon_1 + \epsilon_2)\}. \end{cases}$$

Let $f'$ be a function that minimizes both $\epsilon_1$ and $\epsilon_2$, considering that $\int_{\mathcal{X}} |f'(x)| dx < \infty$, which implies that $f'$ is Lebesgue-integrable on $\mathcal{X}$. The $\mathcal{H}$ represent the fully-connected ReLU networks with width $d_m \leq n + 4$, where $n$ is the input dimension. According to the Universal Approximation Theorem for Width-Bounded ReLU Networks [30], for any $\epsilon > 0$, there exists a $h \in \mathcal{H}$ such that: $\int_{\mathcal{X}} |h(x) - f'(x)| dx < \epsilon$. Consequently, there exists a hypothesis $h \in \mathcal{H}$ that simultaneously minimizes both $\epsilon_1$ and $\epsilon_2$. leading to the condition $\mathcal{H}_{aux}^* \cap \mathcal{H}_{other}^* \neq \emptyset$. In this case, we have $\min_h(\epsilon_1 + \epsilon_2) = \min_h \epsilon_1 + \min_h \epsilon_2$. We denote $\mathcal{H}_{ood}^* = \mathcal{H}_{aux}^* \cap \mathcal{H}_{other}^*$, thus establishing that $\mathcal{H}_{ood}^* \subset \mathcal{H}_{aux}^*$.

## A.2 Proof of Theorem 1

In this section, we analyze the generalization error of the OOD detector training with auxiliary outliers. First, we recall the setting from Sec. 3.1, our goal is to train a detector with auxiliary outliers that can perform well on real-world OOD data. In other words, we aim to train a model on data sampled from $\mathcal{P}_{\widetilde{\mathcal{X}}} = k_{train}\mathcal{P}_{\mathcal{X}_{id}} + (1 - k_{train})\mathcal{P}_{\mathcal{X}_{aux}}$ to obtain a reliable hypothesis $h$ that can effectively generalize to the unknown test-time distribution $\mathcal{P}_{\mathcal{X}} = k_{test}\mathcal{P}_{\mathcal{X}_{id}} + (1 - k_{test})\mathcal{P}_{\mathcal{X}_{ood}}$.

Next, we develop bounds on the OOD detection performance of a detector training with auxiliary outliers, which can be formulated as follow:

*(Generalization Bound of OOD Detector). Let $\mathcal{D}_{train} = \mathcal{D}_{id} \cup \mathcal{D}_{aux}$, consisting of $M$ samples. For any hypothesis $h \in \mathcal{H}$ and $0 < \delta < 1$, with a probability of at least $1 - \delta$, the following inequality holds:*

$$GError(h) \leq \underbrace{\hat{\epsilon}_{x \sim \mathcal{P}_{\widetilde{\mathcal{X}}}}(h, f)}_{empirical\ error} + \underbrace{\epsilon(h, h_{aux}^*)}_{reducible\ error} + \underbrace{\sup_{h \in \mathcal{H}_{aux}^*} \epsilon_{x \sim \mathcal{P}_{\mathcal{X}}}(h, h_{ood}^*)}_{distribution\ shift\ error} + \underbrace{\mathcal{R}_m(\mathcal{H})}_{complexity} + \sqrt{\frac{\ln(\frac{1}{\delta})}{2M}} + \beta, \quad (16)$$

where $\hat{\epsilon}_{x \sim \mathcal{P}_{\widetilde{\mathcal{X}}}}(h, f)$ is the empirical error. We define $\epsilon(h, h_{aux}^*) = \int |\phi_{\mathcal{X}}(x) - \phi_{\widetilde{\mathcal{X}}}(x)||h(x) - h_{aux}^*(x)|dx$ is the reducible error, $\phi_{\mathcal{X}}$ and $\phi_{\widetilde{\mathcal{X}}}$ is the density function of $\mathcal{P}_{\mathcal{X}}$ and $\mathcal{P}_{\widetilde{\mathcal{X}}}$ respectively. $\sup_{h \in \mathcal{H}_{aux}^*} \epsilon_{x \sim \mathcal{P}_{\mathcal{X}}}(h, h_{ood}^*)$ is the distribution shift color, $\mathcal{R}_m(\mathcal{H})$ represents the Rademacher complexity, $\beta$ is the error related to ideal hypotheses. The roadmap of our analysis is as follows:

**Roadmap.** We first show how to bound the OOD detection error in terms of the generalization error on $\mathcal{P}_{\widetilde{\mathcal{X}}}$ and the maximum distribution shift error as well as the reducible error which can be reduced to a small value as the model is optimized. Then, we study the generalization bound from the perspective of Rademacher complexity. We use complexity-based learning theory to quantify the generalization error on $\mathcal{P}_{\widetilde{\mathcal{X}}}$. In the end, we bound the OOD detection generalization error in terms of the empirical error on the training data, the reducible error, the maximum distribution shift error, and the complexity. We also provide detailed proof steps as follows:

**Proof.** This proof relies on the triangle inequality for classification error [4; 9], which implies that for any labeling functions $f_1$, $f_2$, and $f_3$, we have $\epsilon(f_1, f_2) \leq \epsilon(f_1, f_3) + \epsilon(f_2, f_3)$.

$$GError(h) = \epsilon_{x \sim \mathcal{P}_{\mathcal{X}}}(h, f)$$
$$\leq \epsilon_{x \sim \mathcal{P}_{\mathcal{X}}}(h, h_{ood}^*) + \epsilon_{x \sim \mathcal{P}_{\mathcal{X}}}(h_{ood}^*, f)$$
$$= \epsilon_{x \sim \mathcal{P}_{\mathcal{X}}}(h, h_{ood}^*) + \epsilon_{x \sim \mathcal{P}_{\mathcal{X}}}(h_{ood}^*, f) + \epsilon_{x \sim \mathcal{P}_{\widetilde{\mathcal{X}}}}(h, h_{ood}^*) - \epsilon_{x \sim \mathcal{P}_{\mathcal{X}}}(h, h_{ood}^*)$$
$$= \epsilon_{x \sim \mathcal{P}_{\widetilde{\mathcal{X}}}}(h, h_{ood}^*) + \epsilon_{x \sim \mathcal{P}_{\mathcal{X}}}(h_{ood}^*, f) + \epsilon_{x \sim \mathcal{P}_{\mathcal{X}}}(h, h_{ood}^*) - \epsilon_{x \sim \mathcal{P}_{\widetilde{\mathcal{X}}}}(h, h_{ood}^*)$$
$$\leq \epsilon_{x \sim \mathcal{P}_{\widetilde{\mathcal{X}}}}(h, f) + \epsilon_{x \sim \mathcal{P}_{\widetilde{\mathcal{X}}}}(h_{ood}^*, f) + \epsilon_{x \sim \mathcal{P}_{\mathcal{X}}}(h_{ood}^*, f) + \epsilon_{x \sim \mathcal{P}_{\mathcal{X}}}(h, h_{ood}^*) - \epsilon_{x \sim \mathcal{P}_{\widetilde{\mathcal{X}}}}(h, h_{ood}^*)$$

Let $\phi_{\mathcal{X}}$ and $\phi_{\widetilde{\mathcal{X}}}$ be the density functions of $\mathcal{P}_{\mathcal{X}}$ and $\mathcal{P}_{\widetilde{\mathcal{X}}}$, respectively.

$$GError(h) \leq \epsilon_{x \sim \mathcal{P}_{\widetilde{\mathcal{X}}}}(h, f) + \epsilon_{x \sim \mathcal{P}_{\widetilde{\mathcal{X}}}}(h_{ood}^*, f) + \epsilon_{x \sim \mathcal{P}_{\mathcal{X}}}(h_{ood}^*, f)$$

$$+ \int \phi_{\mathcal{X}}(x)|h(x) - h_{ood}^*(x)| \, dx - \int \phi_{\widetilde{\mathcal{X}}}(x)|h(x) - h_{ood}^*(x)| \, dx$$

$$\leq \epsilon_{x \sim \mathcal{P}_{\widetilde{\mathcal{X}}}}(h, f) + \epsilon_{x \sim \mathcal{P}_{\widetilde{\mathcal{X}}}}(h_{ood}^*, f) + \epsilon_{x \sim \mathcal{P}_{\mathcal{X}}}(h_{ood}^*, f) + \int |\phi_{\mathcal{X}}(x) - \phi_{\widetilde{\mathcal{X}}}(x)| \, |h(x) - h_{ood}^*(x)| \, dx$$

$$\leq \epsilon_{x \sim \mathcal{P}_{\widetilde{\mathcal{X}}}}(h, f) + \epsilon_{x \sim \mathcal{P}_{\widetilde{\mathcal{X}}}}(h_{ood}^*, f) + \epsilon_{x \sim \mathcal{P}_{\mathcal{X}}}(h_{ood}^*, f) + \int |\phi_{\mathcal{X}}(x) - \phi_{\widetilde{\mathcal{X}}}(x)| \, |h(x) - h_{aux}^*(x)| \, dx$$

$$+ \int |\phi_{\mathcal{X}}(x) - \phi_{\widetilde{\mathcal{X}}}(x)| \, |h_{aux}^*(x) - h_{ood}^*(x)| \, dx$$

$$\leq \epsilon_{x \sim \mathcal{P}_{\widetilde{\mathcal{X}}}}(h, f) + \epsilon_{x \sim \mathcal{P}_{\widetilde{\mathcal{X}}}}(h_{ood}^*, f) + \epsilon_{x \sim \mathcal{P}_{\mathcal{X}}}(h_{ood}^*, f) + \int |\phi_{\mathcal{X}}(x) - \phi_{\widetilde{\mathcal{X}}}(x)| \, |h(x) - h_{aux}^*(x)| \, dx$$

$$+ \int \phi_{\mathcal{X}}(x) \, |h_{aux}^*(x) - h_{ood}^*(x)| \, dx + \int \phi_{\widetilde{\mathcal{X}}}(x) \, |h_{aux}^*(x) - h_{ood}^*(x)| \, dx$$

$$= \epsilon_{x \sim \mathcal{P}_{\widetilde{\mathcal{X}}}}(h, f) + \epsilon_{x \sim \mathcal{P}_{\widetilde{\mathcal{X}}}}(h_{ood}^*, f) + \epsilon_{x \sim \mathcal{P}_{\mathcal{X}}}(h_{ood}^*, f) + \int |\phi_{\mathcal{X}}(x) - \phi_{\widetilde{\mathcal{X}}}(x)| \, |h(x) - h_{aux}^*(x)| \, dx$$

$$+ \epsilon_{x \sim \mathcal{P}_{\mathcal{X}}}(h_{aux}^*, h_{ood}^*) + \epsilon_{x \sim \mathcal{P}_{\widetilde{\mathcal{X}}}}(h_{aux}^*, h_{ood}^*)$$

$$\leq \epsilon_{x \sim \mathcal{P}_{\widetilde{\mathcal{X}}}}(h, f) + \epsilon_{x \sim \mathcal{P}_{\widetilde{\mathcal{X}}}}(h_{ood}^*, f) + \epsilon_{x \sim \mathcal{P}_{\mathcal{X}}}(h_{ood}^*, f) + \int |\phi_{\mathcal{X}}(x) - \phi_{\widetilde{\mathcal{X}}}(x)| \, |h(x) - h_{aux}^*(x)| \, dx$$

$$+ \epsilon_{x \sim \mathcal{P}_{\mathcal{X}}}(h_{aux}^*, h_{ood}^*) + \epsilon_{x \sim \mathcal{P}_{\widetilde{\mathcal{X}}}}(h_{aux}^*, f) + \epsilon_{x \sim \mathcal{P}_{\widetilde{\mathcal{X}}}}(h_{ood}^*, f),$$

Given that $\min_{h \in \mathcal{H}} \epsilon_{x \sim \mathcal{P}_{\mathcal{X}}}(h, f)$ and $\min_{h \in \mathcal{H}} \epsilon_{x \sim \mathcal{P}_{\widetilde{\mathcal{X}}}}(h, f)$ represent the error of $h_{ood}^*$ and $h_{aux}^*$ on distributions $\mathcal{P}_{\mathcal{X}}$ and $\mathcal{P}_{\widetilde{\mathcal{X}}}$, respectively, we have $\epsilon_{x \sim \mathcal{P}_{\mathcal{X}}}(h_{ood}^*, f) = \min_{h \in \mathcal{H}} \epsilon_{x \sim \mathcal{P}_{\mathcal{X}}}(h, f)$ and $\epsilon_{x \sim \mathcal{P}_{\widetilde{\mathcal{X}}}}(h_{aux}^*, f) = \min_{h \in \mathcal{H}} \epsilon_{x \sim \mathcal{P}_{\widetilde{\mathcal{X}}}}(h, f)$. Considering that $\mathcal{H}_{ood}^* \subset \mathcal{H}_{aux}^*$, it follows that for any $h \in \mathcal{H}_{ood}^*$, $h \in \mathcal{H}_{aux}^*$ is holds. As a result, we have $\epsilon_{x \sim \mathcal{P}_{\widetilde{\mathcal{X}}}}(h_{ood}^*, f) = \min_{h \in \mathcal{H}} \epsilon_{x \sim \mathcal{P}_{\widetilde{\mathcal{X}}}}(h, f)$. Thus, we obtain the following:

$$GError(h) \leq \epsilon_{x \sim \mathcal{P}_{\widetilde{\mathcal{X}}}}(h, f) + \min_{h \in \mathcal{H}} \epsilon_{x \sim \mathcal{P}_{\widetilde{\mathcal{X}}}}(h, f) + \min_{h \in \mathcal{H}} \epsilon_{x \sim \mathcal{P}_{\mathcal{X}}}(h, f)$$

$$+ \int |\phi_{\mathcal{X}}(x) - \phi_{\widetilde{\mathcal{X}}}(x)| \, |h(x) - h_{aux}^*(x)| \, dx + \epsilon_{x \sim \mathcal{P}_{\mathcal{X}}}(h_{aux}^*, h_{ood}^*)$$

$$+ \min_{h \in \mathcal{H}} \epsilon_{x \sim \mathcal{P}_{\widetilde{\mathcal{X}}}}(h, f) + \min_{h \in \mathcal{H}} \epsilon_{x \sim \mathcal{P}_{\widetilde{\mathcal{X}}}}(h, f),$$

We can demonstrate that $\min_{h \in \mathcal{H}} \epsilon_{x \sim \mathcal{P}_{\mathcal{X}}}(h, f) \geq \min_{h \in \mathcal{H}} \epsilon_{x \sim \mathcal{P}_{\widetilde{\mathcal{X}}}}(h, f)$ as follows:

$$\min_{h \in \mathcal{H}} \epsilon_{x \sim \mathcal{P}_{\mathcal{X}}}(h, f) = \min_{h \in \mathcal{H}} \int_{\mathcal{X}} |h(x) - f(x)| dx$$

$$= \min_{h \in \mathcal{H}} \left( \int_{\widetilde{\mathcal{X}}} |h(x) - f(x)| dx + \int_{\mathcal{X} \setminus \widetilde{\mathcal{X}}} |h(x) - f(x)| dx \right)$$

$$\geq \min_{h \in \mathcal{H}} \int_{\widetilde{\mathcal{X}}} |h(x) - f(x)| dx + \min_{h \in \mathcal{H}} \int_{\mathcal{X} \setminus \widetilde{\mathcal{X}}} |h(x) - f(x)| dx$$

$$\geq \min_{h \in \mathcal{H}} \int_{\widetilde{\mathcal{X}}} |h(x) - f(x)| dx$$

$$= \min_{h \in \mathcal{H}} \epsilon_{x \sim \mathcal{P}_{\widetilde{\mathcal{X}}}}(h, f).$$

Thus, We obtain:

$$GError(h) \leq \epsilon_{x \sim \mathcal{P}_{\widetilde{\mathcal{X}}}}(h, f) + \int |\phi_{\mathcal{X}}(x) - \phi_{\widetilde{\mathcal{X}}}(x)| \, |h(x) - h_{aux}^*(x)| \, dx + \epsilon_{x \sim \mathcal{P}_{\mathcal{X}}}(h_{aux}^*, h_{ood}^*)$$

$$+ 4 \min_{h \in \mathcal{H}} \epsilon_{x \sim \mathcal{P}_{\mathcal{X}}}(h, f),$$

We denote $\beta = 4 \min\limits_{h \in \mathcal{H}} \epsilon_{x \sim \mathcal{P}_{\mathcal{X}}}(h, f)$, so

$$GError(h) \leq \epsilon_{x \sim \mathcal{P}_{\widetilde{\mathcal{X}}}}(h, f) + \int |\phi_{\mathcal{X}}(x) - \phi_{\widetilde{\mathcal{X}}}(x)||h(x) - h^*_{aux}(x)| \, dx + \epsilon_{x \sim \mathcal{P}_{\mathcal{X}}}(h^*_{aux}, h^*_{ood}) + \beta,$$

Consider an upper bound on the distribution shift error $\epsilon_{x \sim \mathcal{P}_{\mathcal{X}}}(h^*_{aux}, h^*_{ood})$

$$GError(h) \leq \epsilon_{x \sim \mathcal{P}_{\widetilde{\mathcal{X}}}}(h, f) + \int |\phi_{\mathcal{X}}(x) - \phi_{\widetilde{\mathcal{X}}}(x)||h(x) - h^*_{aux}(x)| \, dx$$
$$+ \sup_{h \in \mathcal{H}^*_{aux}} \epsilon_{x \sim \mathcal{P}_{\mathcal{X}}}(h, h^*_{ood}) + \beta,$$

Next, we recap the Rademacher complexity measure for model complexity. We use complexity-based learning theory [3] (Theorem 8) to quantify the generalization error. Let $\mathcal{D}_{train} = \mathcal{D}_{id} \cup \mathcal{D}_{aux}$ consisting of $M$ samples, $\hat{\epsilon}_{x \sim \mathcal{P}_{\widetilde{\mathcal{X}}}}(h, f)$ is the empirical error of $h$. Then for any hypothesis $h$ in $\mathcal{H}$ (i.e., $\mathcal{H} : \mathcal{X} \rightarrow \{0, 1\}, h \in \mathcal{H}$) and $1 > \delta > 0$, with probability at least $1 - \delta$, we have

$$\epsilon_{x \sim \mathcal{P}_{\widetilde{\mathcal{X}}}}(h, f) \leq \hat{\epsilon}_{x \sim \mathcal{P}_{\widetilde{\mathcal{X}}}}(h, f) + \mathcal{R}_m(\mathcal{H}) + \sqrt{\frac{ln(\frac{1}{\delta})}{2M}}$$

where $\mathcal{R}_m(\mathcal{H})$ is the Rademacher complexities. Finally, it holds with a probability of at least $1 - \delta$ that

$$\epsilon_{x \sim \mathcal{P}_{\mathcal{X}}}(h, f) \leq \underbrace{\hat{\epsilon}_{x \sim \mathcal{P}_{\widetilde{\mathcal{X}}}}(h, f)}_{\text{empirical error}} + \underbrace{\epsilon(h, h^*_{aux})}_{\text{reducible error}} + \underbrace{\sup_{h \in \mathcal{H}^*_{aux}} \epsilon_{x \sim \mathcal{P}_{\mathcal{X}}}(h, h^*_{ood})}_{\text{distribution shift error}} + \underbrace{\mathcal{R}_m(\mathcal{H})}_{\text{complexity}} + \sqrt{\frac{ln(\frac{1}{\delta})}{2M}} + \beta$$

where $\epsilon(h, h^*_{aux}) = \int |\phi_{\mathcal{X}}(x) - \phi_{\widetilde{\mathcal{X}}}(x)| |h(x) - h^*_{aux}(x)| \, dx$ represents the reducible error and $\beta$ is the error related to ideal hypotheses. Notably, when $\beta$ is large, there exists no detector that performs well on $\mathcal{P}_{\mathcal{X}}$, making it unfeasible to find a good hypothesis through training with auxiliary outliers.

### A.3 Proof of Theorem 2

In this section, we proof that diverse outliers enhance generalization, which can be formulated as follows:

*Let $\mathcal{O}(GError(h))$ and $\mathcal{O}(GError(h_{div}))$ represent the upper bounds of the generalization error of detector training with vanilla auxiliary outliers $\mathcal{D}_{aux}$ and diverse auxiliary outliers $\mathcal{D}_{div}$, respectively. For any hypothesis $h$ and $h_{div}$ in $\mathcal{H}$, and $0 < \delta < 1$, with a probability of at least $1 - \delta$, the following inequality holds*

$$\mathcal{O}(GError(h_{div})) \leq \mathcal{O}(GError(h)). \tag{17}$$

The detailed proof proceeds as follows:

**Proof.** At first, we prove that diverse outliers correspond to a smaller distribution shift error than vanilla outliers. Because $\mathcal{X}_{aux} \subset \mathcal{X}_{div}$ holds, the hypotheses performing well on $\mathcal{P}_{\mathcal{X}_{div}}$ also perform well on $\mathcal{P}_{\mathcal{X}_{aux}}$, giving rise to $\mathcal{H}^*_{div} \subset \mathcal{H}^*_{aux}$.

$$\sup_{h \in \mathcal{H}^*_{div}} \epsilon_{x \sim \mathcal{P}_{\mathcal{X}}}(h, h^*_{ood}) \leq \max\{ \sup_{h \in \mathcal{H}^*_{div}} \epsilon_{x \sim \mathcal{P}_{\mathcal{X}}}(h, h^*_{ood}), \sup_{h \in \mathcal{H}^*_{aux} - \mathcal{H}^*_{div}} \epsilon_{x \sim \mathcal{P}_{\mathcal{X}}}(h, h^*_{ood})\},$$

note that
$$\max\{ \sup_{h \in \mathcal{H}^*_{div}} \epsilon_{x \sim \mathcal{P}_{\mathcal{X}}}(h, h^*_{ood}), \sup_{h \in \mathcal{H}^*_{aux} - \mathcal{H}^*_{div}} \epsilon_{x \sim \mathcal{P}_{\mathcal{X}}}(h, h^*_{ood})\} = \sup_{h \in \mathcal{H}^*_{aux}} \epsilon_{x \sim \mathcal{P}_{\mathcal{X}}}(h, h^*_{ood}),$$

Consequently, we have

$$\sup_{h \in \mathcal{H}^*_{div}} \epsilon_{x \sim \mathcal{P}_{\mathcal{X}}}(h, h^*_{ood}) \leq \sup_{h \in \mathcal{H}^*_{aux}} \epsilon_{x \sim \mathcal{P}_{\mathcal{X}}}(h, h^*_{ood}). \tag{18}$$

Furthermore, model effective training leads to small empirical error and small reducible error, if we continue to use the same model architecture, the intrinsic complexity of the model $\mathcal{R}_m(\mathcal{H})$ remains invariant, consider that $\beta$ is a small constant value, therefore, it holds that

$$\mathcal{O}(GError(h_{div})) \leq \mathcal{O}(GError(h)), \tag{19}$$

with a probability of at least $1 - \delta$.

## A.4 Proof of Lemma 1

In this section, we give the proof of the Lemma 1, which can be formalized as follow:

**(Diversity Enhancement with Mixup).** *For a group of mixup transforms[4] $\mathcal{G}$ acting on the input space $\mathcal{X}_{aux}$ to generate an augmented input space $\mathcal{G}\mathcal{X}_{aux}$, defined as $\mathcal{G}\mathcal{X}_{aux} = \{\hat{x} | \hat{x} = \lambda x_1 + (1 - \lambda)x_2; x_1, x_2 \in \mathcal{X}_{aux}, \lambda \in [0, 1]\}$, the following relation holds:*

$$\mathcal{X}_{aux} \subset \mathcal{G}\mathcal{X}_{aux}. \tag{20}$$

**Proof.** $\mathcal{X}_{aux} = \mathcal{X}_{aux}^{y_1} \cup \ldots \cup \mathcal{X}_{aux}^{y_i} \cup \ldots \cup \mathcal{X}_{aux}^{y_j} \cup \ldots \cup \mathcal{X}_{aux}^{y_n}$. Consider performing mixup to obtain a mixed outlier $\hat{x} = \lambda x_i + (1 - \lambda)x_j$, where $x_i \in \mathcal{X}_{aux}^{y_i}, x_j \in \mathcal{X}_{aux}^{y_j}$ and $y_i \neq y_j$. According to assumption 1, there exists $\lambda$ such that $\hat{x}$ exhibits different semantics from the original, i.e., $\hat{x} \notin \mathcal{X}_{aux}^{y_i}$ and $\hat{x} \notin \mathcal{X}_{aux}^{y_j}$. Clearly, the semantic of $\hat{x}$ is also inconsistent with other outliers in $\mathcal{X}_{aux}$. Therefore, $\hat{x} \notin \mathcal{X}_{aux}$. We define $\mathcal{X}_{mix} = \{\hat{x} \mid \hat{x} \notin \mathcal{X}_{aux}, \hat{x} = \lambda x_i + (1 - \lambda)x_j, x_i, x_j \in \mathcal{X}_{aux}\}$ to represents the input space of mixed outliers with distinct semantic to the original. Consequently, $\mathcal{G}\mathcal{X}_{aux} = \mathcal{X}_{aux} \cup \mathcal{X}_{mix}$, leading to $\mathcal{G}\mathcal{X}_{aux} \supset \mathcal{X}_{aux}$.

# B  Experimental Details

## B.1  Details of Dataset

**Auxiliary OOD datasets.** ∘ For CIFAR experiments, we employ the downsampled ImageNet dataset (*ImageNet* $64 \times 64$) as a variant of the original ImageNet dataset, comprising 1,281,167 images with dimensions of 64×64 pixels and organized into 1000 distinct classes. Notably, there is overlap between some of these classes and those present in *CIFAR-10* and *CIFAR-100* datasets. It is important to emphasize that we abstain from utilizing any label information from this dataset, thereby regarding it as an unlabeled auxiliary OOD dataset. To augment the dataset, we apply a random cropping procedure to the 64×64 images, resulting in 32×32 pixel images with a 4-pixel padding. This operation performed with a high probability ensures that the resulting images are unlikely to contain objects corresponding to the ID classes, even if the original images featured such objects. Consequently, we retain a substantial quantity of OOD data for training purposes, yielding a low proportion of ID data within the auxiliary outliers. For conciseness and clarity, we refer to this dataset as *ImageNet-RC*.∘ For ImageNet experiments, we selected a subset from *ImageNet-1K*, which includes 200 classes, to serve as the ID data. Images from the other 800 classes are used as auxiliary datasets, following the setting of [50]. The resolution for both ID and auxiliary images are $224 \times 224$.

**Test OOD datasets.** For CIFAR experiments, we follow the setting in [6; 34]. Specifically, we employ six different natural image datasets as our OOD test datasets, while *CIFAR-10* and *CIFAR-100* serve as our ID test datasets. These six datasets are *SVHN* [37], *Textures* [8], *Places365* [57], *LSUN (crop)*, *LSUN (resize)* [53], and *iSUN* [49]. Below, we provide detailed information about these OOD test datasets, all of which consist of $32 \times 32$ pixel images. ∘ **SVHN.** The *SVHN* dataset [37] comprises color images of house numbers, encompassing ten different digit classes from 0 to 9. For our evaluation, we randomly select 1,000 test images from each digit class, creating a new test dataset with 10,000 images. ∘ **Textures.** The Describable Textures Dataset [8] consists of textural images in the wild. We include the entire collection of 5,640 images for evaluation. ∘ **Places365.** The *Places365* dataset [57] comprises a large-scale photographs depicting scenes classified into 365 scene categories. In the test set, there are 900 images per category. We randomly sample 10,000 images from the test set for our evaluation. ∘ **LSUN (crop)** and **LSUN (resize)**. The Large-scale Scene Understanding dataset *(LSUN)* [53] offers a testing set containing 10,000 images from 10 different scenes. We create two variants of this dataset, namely *LSUN (crop)* and *LSUN (resize)*. *LSUN (crop)* is generated by randomly cropping image patches to the size of $32 \times 32$ pixels, while *LSUN (resize)* involves downsampling each image to the same size. ∘ **iSUN.** The *iSUN* dataset [49] is a subset of *SUN* images. We incorporate the entire collection of 8,925 images from *iSUN* for our evaluation.

In ImageNet experiments, we follow the settings of [50], where *OpenImage-O* [44], *SSB-hard* [43], *Textures* [8], *iNaturalist* [42] and *NINCO* [5] are selected as OOD test datasets. We include *SSB-hard* and *NINCO* in the near-OOD group, while the far-OOD group considers *iNaturalist*, *Textures*, and *OpenImage-O*. ∘ **OpenImage-O** contains 17632 manually filtered images and is $7.8 \times$ larger than

---

[4]The set of all possible combinations of data points and all possible values of $\lambda$ for mixup.

Table 4: **Main results with standard deviation.** Comparison with competitive OOD detection methods trained with the same DenseNet backbone. The performance metrics are averaged (%) over six OOD test datasets from Section 5.1. Some baseline results are sourced from [34]. The best results are in **bold**. *diverseMix not only demonstrates state-of-the-art OOD detection performance on the CIFAR benchmark but also maintains high accuracy in ID classification.*

| Method | CIFAR-10 | | | | CIFAR-100 | | | | w./w.o. $\mathcal{D}_{aux}$ |
|---|---|---|---|---|---|---|---|---|---|
| | FPR95 ($\downarrow$) | AUROC ($\uparrow$) | AUPR ($\uparrow$) | ID-ACC | FPR95 ($\downarrow$) | AUROC ($\uparrow$) | AUPR ($\uparrow$) | ID-ACC | |
| MSP | 58.98 | 90.63 | 93.18 | 94.39 | 80.30 | 73.13 | 76.97 | 74.05 | $\times$ |
| ODIN | 26.55 | 94.25 | 95.34 | 94.39 | 56.31 | 84.89 | 85.88 | 74.05 | $\times$ |
| Mahalanobis | 29.47 | 89.96 | 89.70 | 94.39 | 47.89 | 85.71 | 87.15 | 74.05 | $\times$ |
| Energy | 28.53 | 94.39 | 95.56 | 94.39 | 65.87 | 81.50 | 84.07 | 74.05 | $\times$ |
| SSD+ | 7.22 | 98.48 | 98.59 | NA | 38.32 | 88.91 | 89.77 | NA | $\times$ |
| OE | 9.66 | 98.34 | 98.55 | 94.12 | 19.54 | 94.93 | 95.26 | 74.25 | $\checkmark$ |
| SOFL | 5.41 | 98.98 | 99.10 | 93.68 | 19.32 | 96.32 | 96.99 | 73.93 | $\checkmark$ |
| CCU | 8.78 | 98.41 | 98.69 | 93.97 | 19.27 | 95.02 | 95.41 | 74.49 | $\checkmark$ |
| Energy (w. $\mathcal{D}_{aux}$) | 4.62 | 98.93 | 99.12 | 92.92 | 19.25 | 96.68 | 97.44 | 72.39 | $\checkmark$ |
| NTOM | $4.00 \pm 0.22$ | $99.09 \pm 0.05$ | $98.61 \pm 0.32$ | $94.26 \pm 0.11$ | $18.77 \pm 0.75$ | $96.69 \pm 0.12$ | $96.49 \pm 0.33$ | $74.52 \pm 0.31$ | $\checkmark$ |
| POEM | $2.54 \pm 0.56$ | $99.40 \pm 0.05$ | $99.50 \pm 0.07$ | $93.49 \pm 0.27$ | $15.14 \pm 1.16$ | $97.79 \pm 0.17$ | $98.31 \pm 0.12$ | $73.41 \pm 0.21$ | $\checkmark$ |
| MixOE | $14.54 \pm 0.87$ | $97.16 \pm 0.17$ | $97.41 \pm 0.16$ | $94.48 \pm 0.09$ | $27.71 \pm 2.22$ | $92.93 \pm 0.85$ | $93.81 \pm 0.69$ | $75.15 \pm 0.14$ | $\checkmark$ |
| DivOE | $11.41 \pm 0.88$ | $97.76 \pm 0.16$ | $98.18 \pm 0.12$ | $94.07 \pm 0.24$ | $18.91 \pm 2.59$ | $95.00 \pm 0.72$ | $95.26 \pm 0.50$ | $74.08 \pm 0.44$ | $\checkmark$ |
| DiverseMix (ours) | $\mathbf{1.92 \pm 0.14}$ | $\mathbf{99.42 \pm 0.01}$ | $\mathbf{99.51 \pm 0.03}$ | $94.16 \pm 0.12$ | $\mathbf{8.51 \pm 0.68}$ | $\mathbf{98.24 \pm 0.10}$ | $\mathbf{98.46 \pm 0.10}$ | $74.60 \pm 0.32$ | $\checkmark$ |

the *ImageNet-O* dataset. ∘ **SSB-hard** is selected from *ImageNet-21K*. It consists of 49K images and covers 980 categories. ∘ *iNaturalist* consists of 859000 images from over 5000 different species of plants and animals. ∘ *NINCO* consists with a total of 5879 samples of 64 classes which are non-overlapped with *ImageNet-1K*.

## B.2 Training Details.

∘ **CIFAR experiments.** We use DenseNet-101 [21] as the backbone for all methods, employing stochastic gradient descent with Nesterov momentum (momentum = 0.9) over 100 epochs. The initial learning rate of 0.1 decreases by a factor of 0.1 at 50, 75, and 90 epochs. Batch sizes are 128 for both ID data and OOD data. For DiverseMix, we set $\alpha = 4$, $T = 10$. Experiments are run over five times to report the means and standard deviations. ∘ **ImageNet experiments.** We use ResNet18 [18] as the backbone network. We use SGD optimizer to train all the models. The momentum is set to 0.9. Model is obtained by training ResNet18 for 100 epochs with an initial learning rate of 0.1, utilizing a cosine annealing strategy to adjust the learning rate. The weight decay is set to 0.0005. Batch size is set to 256 both ID data and OOD data. For DiverseMix, we set $\alpha = 8$, $T = 0.1$. We use OE loss as regularization loss. Experiments are run over five times to report the means and standard deviations.

## B.3 Details of OOD Regularization Method.

In addition to the Energy loss mentioned in Section 4.2, our method can be extended to different OOD regularization methods, such as OE [20] and K+1 [6]. The details are as follows:

**Outlier Exposure (OE).** OE introduces a promising approach towards OOD detection by utilizing outliers to force apart the distributions of ID and OOD. Its scoring function and corresponding regular function can be expressed as:

$$S(x, \theta) = \max \text{softmax}(F(x, \theta)), \quad \mathcal{L}_{aux} = \mathbb{E}_{x \sim \mathcal{D}_{aux}}[\mathcal{L}_{CE}(F(x, \theta), \mathcal{U})], \quad (21)$$

where $\mathcal{U}$ is the uniform distribution over $K$ classes.

**(K+1)-way regularization method.** Considering a (K+1)-way classifier network $F$, where the (K+1)-th label indicates OOD class. Its scoring function and regular function can be expressed as:

$$S(x, \theta) = -\text{softmax}_{K+1}(F(x, \theta)), \quad \mathcal{L}_{aux} = \mathbb{E}_{x \sim \mathcal{D}_{aux}}[\mathcal{L}_{CE}(F(x, \theta), K + 1)], \quad (22)$$

where $\text{softmax}_{K+1}(\cdot)$ represents the softmax output in the K+1 dimension.

## B.4 Details of Main Experiment.

**Full Results with Standard Deviation.** In Tab. 4 and Tab. 5, we present the experimental results for all evaluation metrics along with the corresponding standard deviations. From the experimental results we can draw similar conclusions as those in Sec. 5.

**Results on Individual OOD Dataset.** We also provide the performance of our method on individual OOD dataset in tabel 6.

Table 5: **Main results on large-scale datasets.** Comparison with competitive OOD detection methods trained with the same ResNet backbone. We divide the OOD test set into two distinct groups: near-OOD and far-OOD. For each group, we report the average performance metrics with standard deviation. The best and second-best results are in **bold** and underline respectively. *Echoing the findings from our CIFAR experiments, diverseMix demonstrates strong OOD detection capabilities for both near-OOD and far-OOD test sets, achieving state-of-the-art OOD detection performance.*

| Method | Near-OOD | | | Far-OOD | | | Average | | | ID-ACC |
|---|---|---|---|---|---|---|---|---|---|---|
| | FPR (↓) | AUROC (↑) | AUPR (↑) | FPR (↓) | AUROC (↑) | AUPR (↑) | FPR (↓) | AUROC (↑) | AUPR (↑) | |
| MSP | 70.35±0.67 | 82.75±0.19 | 88.58±0.08 | 54.51±3.30 | 88.81±0.67 | 91.86±0.59 | 60.85±2.23 | 86.39±0.48 | 90.54±0.38 | 85.81±0.14 |
| Energy | 70.35±0.54 | 81.88±0.12 | 88.56±0.04 | 53.87±4.71 | 89.30±0.95 | 91.59±0.74 | 60.46±2.89 | 86.33±0.62 | 90.38±0.45 | 85.81±0.14 |
| Max Logits | 69.45±0.68 | 82.25±0.16 | 88.69±0.04 | 52.49±4.48 | 89.60±0.88 | 92.13±0.67 | 59.28±2.91 | 86.66±0.60 | 90.75±0.42 | 85.81±0.14 |
| ODIN | 69.06±0.62 | 82.20±0.17 | 88.75±0.04 | 50.90±4.33 | 89.90±0.89 | 92.50±0.66 | 58.16±2.78 | 86.82±0.60 | 91.00±0.41 | 85.81±0.14 |
| OE | **59.12±0.57** | **86.86±0.32** | **92.69±0.27** | 54.95±1.47 | 90.51±0.21 | 91.20±0.43 | 56.61±0.94 | 89.05±0.24 | 91.79±0.18 | 85.52±0.18 |
| Energy (w. $D_{aux}$) | 60.67±1.83 | 85.95±0.63 | 91.75±1.03 | 58.07±3.89 | 89.73±0.27 | 89.67±0.56 | 59.11±1.76 | 88.22±0.09 | 90.50±0.08 | 84.94±0.66 |
| DPN | 63.39±1.15 | 84.94±0.07 | 91.46±0.05 | 61.31±2.04 | 89.85±0.31 | 90.16±0.26 | 62.14±1.60 | 87.89±0.21 | 90.68±0.18 | 85.27±0.02 |
| MixOE | 68.43±0.12 | 83.42±0.18 | 88.74±0.24 | 50.51±0.31 | 90.62±0.19 | 92.31±0.14 | 57.68±0.23 | 87.38±0.18 | 90.89±0.18 | 86.35±0.12 |
| DiverseMix (ours) | 59.81±0.28 | 86.36±0.02 | 91.76±0.23 | **48.58±1.51** | **91.35±0.26** | **92.38±0.29** | **53.07±0.80** | **89.36±0.14** | **92.13±0.08** | 85.95±0.13 |

Table 6: **main results on individual OOD dataset.** We provide the results of diverseMix on each individual OOD dataset from Section 5.1. The reported performance of our method is based on five independent training runs using different random seeds.

| OOD dataset | CIFAR-10 | | | | CIFAR-100 | | | |
|---|---|---|---|---|---|---|---|---|
| | FPR (↓) | AUROC (↑) | AUPR (↑) | ID-ACC | FPR (↓) | AUROC (↑) | AUPR (↑) | ID-ACC |
| LSUN-C | 3.39 ± 0.80 | 99.21 ± 0.14 | 99.29 ± 0.13 | 99.41 ± 0.02 | 8.16 ± 2.17 | 98.55 ± 0.35 | 98.65 ± 0.32 | 98.19 ± 0.10 |
| LSUN-R | 0.00 ± 0.00 | 100.00 ± 0.00 | 100.00 ± 0.00 | 99.41 ± 0.02 | 0.01 ± 0.01 | 99.93 ± 0.13 | 99.95 ± 0.08 | 99.83 ± 0.34 |
| ISUN | 0.00 ± 0.00 | 100.00 ± 0.00 | 100.00 ± 0.00 | 99.41 ± 0.02 | 0.04 ± 0.01 | 99.91 ± 0.13 | 99.95 ± 0.07 | 100.00 ± 0.00 |
| DTD | 1.13 ± 0.14 | 99.58 ± 0.06 | 99.72 ± 0.06 | 99.98 ± 0.01 | 5.77 ± 0.61 | 98.47 ± 0.15 | 98.98 ± 0.11 | 99.96 ± 0.01 |
| Places365 | 5.30 ± 0.64 | 98.53 ± 0.17 | 98.60 ± 0.25 | 98.41 ± 0.10 | 26.55 ± 2.45 | 94.95 ± 0.41 | 95.32 ± 0.46 | 93.92 ± 0.42 |
| SVHN | 1.66 ± 0.45 | 99.21 ± 0.18 | 99.39 ± 0.13 | 98.89 ± 0.22 | 10.53 ± 1.36 | 97.60 ± 0.26 | 97.93 ± 0.23 | 96.89 ± 0.39 |
| Average | 1.92 ± 0.14 | 99.42 ± 0.01 | 99.50 ± 0.03 | 99.41 ± 0.02 | 8.51 ± 0.68 | 98.24 ± 0.10 | 98.46 ± 0.10 | 98.19 ± 0.10 |

## B.5 Details of Figure 2.

In this experiment, we aim to explore the effect of outlier quality on OOD detection performance. We analyzed this from two perspectives: the sample size of outliers and the diversity of outliers. Specifically, we constructed a series of subsets from the *Imagenet-RC* dataset to generate low-quality auxiliary outliers datasets with different sample size and diversity. Afterwards, we used these constructed low-quality subsets as the auxiliary outliers dataset to train the model. All experimental results are run over three times and averaged. The experimental details are as follows:

**Decreasing the sample size of auxiliary outliers.** To explore the impact of sample size on our experimental results, we keep the number of classes constant and decrease the size of the auxiliary outliers dataset. This is achieved by applying downsampling techniques, resulting in subsets with the same classes as the original *Imagenet-RC* dataset but with sizes of $\{100\%, 85\%, 70\%, 55\%, 40\%, 25\%, 10\%\}$ compared to the original auxiliary outliers dataset.

**Decreasing the diversity of auxiliary outliers.** To investigate the effect of outlier diversity on OOD detection performance, we further reduce the number of classes included in the subset. Specifically, we keep the sample size of the subset at $10\%$ of the original outliers dataset, but gradually decrease the number of classes included (as the number of classes decreases, the number of samples per class increases, ensuring a consistent overall sample size). We constructed a series of subsets with $\{1000, 850, 700, 550, 400, 250, 100\}$ classes to serve as auxiliary outliers for experimental evaluation.

## B.6 Details of Q5 Ablation Study.

In this section, we conduct an ablation study from two perspectives. Firstly, we compare our method with traditional data augmentation techniques (semantic-preserving) to demonstrate that our method effectively enhances the diversity of outliers by altering their semantics. Secondly, considering that our method is an improved variant of mixup, we investigate different mixup strategies to explore what factors contribute to the performance gains. Detailed experimental results are shown in Table 7.

**Ablation study (I): Ablation study with different data augmentation method.** To investigate if *diverseMix* offers unique advantages over other data augmentation techniques in enhancing the diversity of outliers, we select different data augmentation methods to process the auxiliary outliers and validate their impact on performance. Specifically, we choose semantic-invariant data augmentation methods: *Gaussian noise* [38], *cutout* [11], and *color jitter* for comparison with our method.

Table 7: **Ablation study on different data augmentation methods.** Performance are averaged (%) over six OOD test datasets from Section 5.1. The best results are in **bold**. The reported OOD detection performance is based on five independent training runs using different random seeds.

| Method | CIFAR-10 | | | | CIFAR-100 | | | |
|---|---|---|---|---|---|---|---|---|
| | FPR (↓) | AUROC (↑) | AUPR (↑) | ID-ACC | FPR (↓) | AUROC (↑) | AUPR (↑) | ID-ACC |
| Vanilla | $5.43 \pm 0.18$ | $98.80 \pm 0.06$ | $98.92 \pm 0.04$ | $94.37 \pm 0.10$ | $16.30 \pm 1.74$ | $96.87 \pm 0.44$ | $97.38 \pm 0.36$ | $74.49 \pm 0.29$ |
| Gaussian noise | $6.69 \pm 0.19$ | $98.64 \pm 0.05$ | $98.75 \pm 0.08$ | $94.27 \pm 0.02$ | $19.94 \pm 2.19$ | $95.69 \pm 0.55$ | $96.07 \pm 0.55$ | $74.78 \pm 0.18$ |
| Cutout | $7.20 \pm 0.87$ | $98.57 \pm 0.14$ | $98.72 \pm 0.15$ | $94.17 \pm 0.07$ | $19.12 \pm 0.82$ | $96.64 \pm 0.20$ | $97.18 \pm 0.06$ | $74.88 \pm 0.26$ |
| Color jittering | $4.34 \pm 0.72$ | $99.04 \pm 0.13$ | $99.09 \pm 0.16$ | $94.22 \pm 0.06$ | $14.47 \pm 1.75$ | $97.02 \pm 0.39$ | $97.33 \pm 0.36$ | $74.46 \pm 0.20$ |
| Mixup | $4.00 \pm 0.12$ | $99.02 \pm 0.06$ | $99.07 \pm 0.09$ | $94.16 \pm 0.08$ | $12.56 \pm 0.30$ | $97.42 \pm 0.09$ | $97.63 \pm 0.05$ | $74.76 \pm 0.19$ |
| Cutmix | $6.38 \pm 0.75$ | $98.65 \pm 0.19$ | $98.80 \pm 0.18$ | $94.10 \pm 0.16$ | $17.07 \pm 0.97$ | $96.87 \pm 0.17$ | $97.30 \pm 0.17$ | $74.80 \pm 0.19$ |
| DiverseMix (ours) | $\mathbf{1.92 \pm 0.14}$ | $\mathbf{99.42 \pm 0.01}$ | $\mathbf{99.50 \pm 0.03}$ | $94.16 \pm 0.12$ | $\mathbf{8.51 \pm 0.68}$ | $\mathbf{98.24 \pm 0.10}$ | $\mathbf{98.46 \pm 0.10}$ | $74.60 \pm 0.32$ |

**Gaussian noise.** Here, we introduce an appropriate level of noise to the training data to augment its diversity and quantity. We incorporate Gaussian noise with a mean of 0 and a variance of 0.1. To effectively mitigate the risk of model overfitting to Gaussian noise, wherein the model incorrectly classifies any image with Gaussian noise as an OOD input and any noise-free image as an ID sample, this type of noise is applied to only half of the outlier samples during the model training phase.

**Cutout.** Cutout is a data augmentation technique that introduces random masking of small regions in input images, preventing the model from relying on specific features. In our study, we apply the cutout augmentation to half of the auxiliary outlier samples. This involves randomly masking out small regions within these outlier images by setting all pixel values in the masked regions to zero.

**Color jittering.** Color jittering is a widely adopted data augmentation technique in image processing. It introduces random variations to the brightness, contrast, saturation, and hue of an image, simulating the diverse conditions encountered in real-world scenarios, such as different lighting environments or camera settings. Specifically, for each auxiliary outlier image, we randomly adjust its brightness within a range of $\pm 0.4$, its contrast within a range of $\pm 0.4$ and its saturation within a range of $\pm 0.4$, while rotating the hue by $\pm 0.1$ radians. This data augmentation strategy preserves the semantic content of the original outlier image while introducing controlled variations in color properties.

**Ablation study (II): Ablation study with different semantic interpolation method.** To explore how diverseMix differs from other mixup-based methods, we compared the performance to *vanilla mixup* and *cutmix*. We set the hyperparameter $\alpha = 4$, consistent with our method *diverseMix*.

**Vanilla mixup.** *Vanilla mixup* involves generating virtual training examples (referred to as mixed samples) through linear interpolations between data points and corresponding labels, given by:

$$\hat{x} = \lambda x_i + (1 - \lambda)x_j, \quad \hat{y} = \lambda y_i + (1 - \lambda)y_j, \tag{23}$$

where $(x_i, y_i)$ and $(x_j, y_j)$ are two samples drawn randomly from the empirical training distribution, and $\lambda \in [0, 1]$ is usually sampled from a Beta distribution with parameter $\alpha$ denoted as $Beta(\alpha, \alpha)$.

**Cutmix.** *Cutmix* is a data augmentation method that constructs virtual training examples by performing cutting and replacing the cutted region with the corresponding region from the other image:

$$\hat{x} = M(\lambda) \odot x_i + (1 - M(\lambda)) \odot x_j, \quad \hat{y} = \lambda y_i + (1 - \lambda)y_j, \tag{24}$$

where $M(\lambda)$ is a binary mask randomly chosen covering $\lambda$ proportion of the input, and $\odot$ represents the element-wise product. Here, $\lambda$ is usually sampled from a preset beta distribution $Beta(\alpha, \alpha)$.

## C   Additional Results

### C.1   Hyperparameter Analysis.

In this section, we analyze the main hyperparameters involved in our method. The experimental results are shown in the *table 8*. From the experimental results, we find that diverseMix is more effective with larger values of $\alpha$. A larger $\alpha$ means that the model will adopt a more aggressive interpolation strategy, generating mixed outliers that deviates further from the original samples. This aligns with our expectations. The temperature $T$ controls diverseMix's sensitivity to the samples, an appropriate $T$ allows diverseMix to accurately perceive the model's familiarity with the samples. $\omega$ controls the strength of regularization, an excessively large $\omega$ may impair the classification performance. In addition, we provide our strategy for hyperparameter adjustment in practice as follows:

**hyper-parameter tuning.** We can first determine the largest possible value of $\omega$ for the original baseline model while maintaining the ID classification accuracy. Then, we can select more suitable

Table 8: **Hyperparameter analysis.** Performance averaged (%) over six OOD test datasets from Section 5.1. The performance reported are averaged over different random seeds.

| $\alpha$ | $T$ | $\omega$ | CIFAR-10 | | | | CIFAR-100 | | | |
|---|---|---|---|---|---|---|---|---|---|---|
| | | | FPR ($\downarrow$) | AUROC ($\uparrow$) | AUPR ($\uparrow$) | ID-ACC | FPR ($\downarrow$) | AUROC ($\uparrow$) | AUPR ($\uparrow$) | ID-ACC |
| 0.5 | 10 | 0.01 | 3.67±0.41 | 99.15±0.06 | 99.24±0.06 | 94.19±0.08 | 9.39±1.47 | 98.03±0.21 | 98.23±0.19 | 74.67±0.20 |
| 1 | 10 | 0.01 | 3.16±0.18 | 99.23±0.03 | 99.33±0.04 | 94.13±0.08 | 9.55±0.99 | 98.10±0.16 | 98.36±0.19 | 74.66±0.10 |
| 2 | 10 | 0.01 | 2.55±0.68 | 99.32±0.10 | 99.41±0.11 | 94.20±0.05 | 8.44±0.51 | 98.17±0.05 | 98.41±0.04 | 74.70±0.22 |
| 4 | 5 | 0.01 | 3.87±0.31 | 99.08±0.05 | 99.19±0.01 | 94.42±0.09 | 10.27±0.97 | 98.04±0.16 | 98.29±0.11 | 74.67±0.33 |
| 4 | 1 | 0.01 | 5.43±0.59 | 98.78±0.16 | 98.93±0.16 | 94.01±0.22 | 15.62±1.71 | 97.18±0.29 | 97.63±0.20 | 74.90±0.12 |
| 4 | 20 | 0.01 | 2.75±0.38 | 99.26±0.11 | 99.33±0.12 | 94.15±0.06 | 9.49±0.98 | 98.04±0.03 | 98.29±0.04 | 74.43±0.39 |
| 4 | 10 | 0.05 | 1.76±0.02 | 99.47±0.02 | 99.56±0.02 | 93.89±0.15 | 8.12±1.03 | 98.36±0.17 | 98.58±0.14 | 73.94±0.28 |
| 4 | 10 | 0.1 | 2.60±0.62 | 99.33±0.09 | 99.45±0.07 | 92.51±0.44 | 9.04±1.00 | 98.19±0.08 | 98.45±0.05 | 71.96±0.69 |
| 4 | 10 | 0.01 | 1.92 ± 0.14 | 99.42 ± 0.01 | 99.50 ± 0.03 | 94.16 ± 0.12 | 8.51 ± 0.68 | 98.24 ± 0.10 | 98.46 ± 0.10 | 74.60±0.32 |

Table 9: **Experimental Results on NLP OOD detection task.** The best results are in **bold**. The same network architecture is used for all three detectors. All results are represented in percentages. *Our method diverseMix also achieves good performance in the field of natural language proceeding.*

| OOD testset | FPR95 ($\downarrow$) | | | AUROC ($\uparrow$) | | | AUPR ($\uparrow$) | | |
|---|---|---|---|---|---|---|---|---|---|
| | MSP | OE | diverseMix | MSP | OE | diverseMix | MSP | OE | diverseMix |
| SNLI | 52.61 | 27.05±2.23 | **21.58±4.51** | 76.19 | 87.63±1.03 | **90.12±0.66** | 33.83 | 50.80±3.21 | **58.45±1.27** |
| Multi30k | 76.00 | 35.69±2.90 | **19.38±3.83** | 60.69 | 87.09±1.48 | **92.43±1.05** | 22.08 | 56.93±4.24 | **68.37±5.53** |
| WMT16 | 68.66 | 14.28±1.70 | **11.90±3.01** | 67.30 | 94.59±0.44 | **95.59±0.67** | 26.36 | 75.65±2.04 | **79.80±5.03** |
| Yelp Reviews | 82.98 | 5.36±0.71 | **4.23±2.96** | 56.38 | 96.98±0.70 | **97.92±0.50** | 20.45 | 78.09±6.34 | **85.09±5.87** |
| Average | 70.06 | 20.60±1.58 | **14.27±2.56** | 65.14 | 91.57±0.72 | **94.02±0.45** | 25.68 | 65.37±3.60 | **72.93±4.82** |

Table 10: **Time and memory cost of different methods.** We compare the computational overhead of DiverseMix and other methods on CIFAR-100 under the same setting. Best results are in **bold**.

| Method | *Computational Cost* | | *OOD Detection Performance* | | | | w./w.o. $\mathcal{D}_{aux}$ |
|---|---|---|---|---|---|---|---|
| | Times (hours) | GPU Memory (MB) | FPR95 ($\downarrow$) | AUROC ($\uparrow$) | AUPR ($\uparrow$) | ID-ACC | |
| Energy | **1.99** | **13017** | 19.25 | 96.68 | 97.44 | 72.39 | ✓ |
| NTOM | 2.51 | 15549 | 18.77 | 96.69 | 96.49 | 74.52 | ✓ |
| POEM | 8.47 | 19851 | 15.14 | 97.79 | 98.31 | 73.41 | ✓ |
| DivOE | 4.88 | 13163 | 18.91 | 95.00 | 95.26 | 74.08 | ✓ |
| DiverseMix | 2.20 | 13213 | **8.51** | **98.24** | **98.46** | 74.60 | ✓ |

parameters for $\alpha$ and $T$, with adjustments made using an OOD validation set distinct from the testing OOD dataset. For example, a subset from the auxiliary outliers could serve as an OOD validation set.

## C.2 DiverseMix for OOD Detection in Natural Language Processing.

To further validate the applicability of our method in non-image domains, we explore the use of diverseMix in the task of *Natural Language Processing*, following the setting of OE [20].

**Experimental Setting.** We use the *SST* dataset as the ID data, while utilizing the *WikiText-2* dataset as auxiliary outlier data. We employ the *SNLI*, *Multi30K*, *WMT16*, and *Yelp Reviews* datasets as OOD test set. We use *QRNN* [33] language models as baseline OOD detectors. Initially, we train vanilla models for 50 epochs and subsequently fine-tune them on the *WikiText-2* dataset using *OE* or *DiverseMix* for an additional 5 epochs. Outlier Exposure is implemented by adding the cross entropy to the uniform distribution on tokens from sequences in $\mathcal{D}_{aux}$ as an additional loss term. For *DiverseMix*, we apply mixup strategy at embedding level, and the loss function is consistent with *OE*.

**Experimental Results.** The results presented in table 9 highlight that: 1) The incorporation of auxiliary outliers enhances OOD detection performance in non-image domains. 2) Our method increases the diversity of auxiliary outliers, further enhancing the model's OOD detection performance.

## C.3 Experiments on Computational Cost.

To better understand the computational budget, we summarize the time and memory cost results in Table 10, which shows that diverseMix can achieve better performance with relatively low time and memory overhead compared with other OOD detection methods that train with auxiliary outliers.

## C.4 Impact Statements

Our work focuses on enhancing AI safety and trustworthiness by improving the robust performance of machine learning models on OOD data, which is crucial for high-stakes tasks in real-world scenarios.

However, biases in benchmark OOD detection data, such as ImageNet, necessitate careful auxiliary outlier selection for safety-critical applications to ensure the proposed method's reliability and safety.

## D   Hardware and Software

We run all the experiments on NVIDIA GeForce RTX 3090 GPU. Our implementations are based on Ubuntu Linux 18.04 with Python 3.8.

