# OpenReview forum: "Out-Of-Distribution Detection with Diversification (Provably)"
_NeurIPS.cc/2024/Conference — NeurIPS 2024 poster_

### Official Review · Reviewer_1vuo · 2024-07-09

**Soundness:** 3
**Presentation:** 3
**Contribution:** 3
**Rating:** 7
**Confidence:** 4

**Summary:**

The authors propose DiverseMixup, a Mixup data augmentation technique applied to auxiliary OOD data to improve the OOD detection capacities of classifiers trained with the Outlier Exposure technique. They provide a theoretical analysis justifying their approach and demonstrate the superior empirical performances of their technique.

**Strengths:**

- The method appears to be novel and competitive compared to other OOD data augmentation methods while remaining quite simple and easy to implement.
- The ablation and complementary experiments are satisfying.
- The experiments answer many practical questions.

**Weaknesses:**

One of the main contributions of the paper is the theoretical analysis. However, there appear to be critical flaws in both the demonstrations and the hypotheses.

### Major

1. The quantities $h^*_{ood}$ and $h^*_{aux}$ are never defined. We can guess out of commonly used notation in optimization that $h^*_{ood} = \\underset{h\\in \\mathcal{H}}{\\operatorname{argmin}} \\epsilon_{P_{\\mathcal{X}}}(h,f)$ (same for $h^*_{aux}$ ) but this is the definition of $\\mathcal{H}^*_{ood}$, which is defined as a set (which is not straightforward - why would an argmin be a set in that case ?). It adds a lot of confusion, and we never know exactly what we are talking about, which is critical for a demonstration.
2. In the demonstration of theorem 1, the authors define $\\beta_1$ and $\\beta_2$ as constants, but 1) they depend on $P_{\\mathcal{X}}$ which is supposed to be affected by the later-introduced $\\mathcal{X_div}$ and 2) the parts of l.530 that are replaced do not seem to match the definition of $\\beta$'s.
3. Demonstrations of Theorems 2 and 3 seem to rely on one argument, which is: "Since $\\mathcal{X_aux} \\subset \\mathcal{X_ood}$, then  $\\mathcal{H}^*_{ood} \\subset \\mathcal{H}^*_{aux}$". I am concerned with the validity of this assumption (assuming that the definition of $\\mathcal{H}^*$ as sets make sense, which is not clear). As a counterexample, let's consider $\\phi_{\\mathcal{X}} = Unif(0,3)$ and $\\phi_{\\tilde{\\mathcal{X}}} = Unif(0,1)$ (which implies that $\\mathcal{X_aux} = (0,1)$ and  $\\mathcal{X_ood} = (0,3)$).  Now, let's consider $h^*_{ood}$ and $h^*_{aux}$ such that

$
\\begin{dcases}
  \\int_{\\mathcal{X_ood} \\setminus \\mathcal{X_aux}} |h^*_{ood}(x) -f(x)|dx = 0\\\\
  \\epsilon_{P_X}(h^*_{ood},f) = \\int_{\\mathcal{X_aux}} |h^*_{ood}(x) -f(x)|dx + \\int_{\\mathcal{X_ood} \\setminus \\mathcal{X_aux}} |h^*_{ood}(x) -f(x)|dx= \\int_{\\mathcal{X_aux}} |h^*_{ood}(x) -f(x)|dx =  \\epsilon_1,\\\\ \\end{dcases}$

and

$\\begin{dcases}
\\epsilon_{P_\\tilde{X}}(h^*_{aux},f) = \\int_{\\mathcal{X_aux}} |h^*_{aux}(x) -f(x)|dx =  \\epsilon_2 < \\epsilon_1,\\\\
  \\int_{\\mathcal{X_aux} \\setminus \\mathcal{X_aux}} |h^*_{aux}(x) -f(x)|dx > \\epsilon_1 - \\epsilon_2
  \\end{dcases}$

  where for simplicity, we omit $\\mathcal{X_id}$, assuming that the behavior is similar on this input space region for $h^*_{ood}$ and $h^*_{aux}$ . In that case, clearly, $h^*_{ood}$ minimizes $\\epsilon_{P_X}(h,f)$ (thanks to the inequality above that keeps $h^*_{aux}$ sub-optimal for $P_X$) but does not minimize $\\epsilon_{P_\\tilde{X}}(h,f)$.

### Minor

1. l. 111 perhaps you meant $\\mathcal{X_id}$ instead of $\\mathcal{Y_id}$?
2. l.122 $\\epsilon$ is called a probability, whereas it is an expectation
3. Some typos.

**Questions:**

I am puzzled because, on the one hand, the paper demonstrates strong empirical results, and the evaluation methodology is extensive and thorough, but on the other hand, I suspect that the authors' theoretical work is flawed - which does not affect the strength of the presented method but the validity of the paper. I am ready to improve my rating to acceptance if the authors prove my suspicions wrong during the rebuttal.

**Limitations:**

The authors have adequately addressed the limitations.

---

> ### Author Rebuttal · Authors · 2024-08-07
>
> We thank the reviewer for recognizing our novel method and satisfying experiments. We appreciate your support and constructive suggestions and address your concerns as follows.
>
> ---
> ## W1. The quantities $h^*_{ood}$ and $h^*_{aux}$ are never defined. Why would an  argmin be a set in that case?
>
> Thank you for your valuable comment. We agree that clearer definitions of $h^*_{ood}$ and $h^*_{aux}$ would improve our manuscript. Our choice to define the argmin as a set was purposeful and well-founded and we appreciate to clarify this as follow:
>
> ### **1) The definition of quantities $h^{*}_{aux}$** **and $h^{*}_{aux}$**.
>
> $h^*_{ood}$ and $h^*_{aux}$ is defined as the element in $\mathcal H^*_{ood}$ and $\mathcal H^*_{aux}$, respectively, which can be denoted as: $h^*_{ood}\in \mathcal H^*_{ood}, h^*_{aux} \in \mathcal H^*_{aux}$, where $\mathcal H^*_{ood}$ and $\mathcal H^*_{aux}$ refers to the sets of ideal hypotheses on the training data distribution $P_{\widetilde{\mathcal{X}}}$ and test-time data distribution $P_{\mathcal{X}}$. We will add the definition in  manuscript.
>
> ### **2) Why would an argmin be a set?**
>
> **(i) Our setting requires us to represent the optimal hypothesis as a set.**
> In our setting, $\mathcal H_{aux}$ contains all hypotheses optimal for the training data distribution. However, these hypotheses may perform inconsistently on the test-time data distribution, necessitating the use of a set rather than a single $h^*_{aux}$.
>
> **(ii) The optimization problem behind deep neural networks is highly non-convex and the optimal solution is not unique [1] [2].** Therefore, defining argmin as a set offers generality.
>
> **(iii) Representing argmin as a set is not unprecedented in the field.** For example, the theoretical analysis in [3] also employs this set-based representation.
>
> ---
> ## W2. The authors define $\beta_1$ and $\beta_2$ as constants, but 1) they depend on  $P_\mathcal X$  which is supposed to be affected by the later-introduced $\mathcal X_{div}$ and 2) the parts of line 530 that are replaced do not seem to match the definition of  $\beta$'s.
>
> Sincerely thank you for your thorough review. We realize our derivation omitted some explanations, leading to misunderstandings.  We're happy to clarify in detail as follow:
>
> ### **1) Clarification of $\beta_1$and $\beta_2$.**
>
> **(i) We do not define $\beta_1$and $\beta_2$ as constants.** This misunderstanding may have arisen from our definition of $\beta$ as a constant in Theorem 1. We will clarify this in the revised paper.
>
> **(ii) $\beta_1$ is not affected by $\mathcal X_{div}$.** $\beta_1$ depends on the unknown test data distribution $P_{\mathcal X}$. The later-introduced $\mathcal X_{div}$ only affects the training data distribution, thus $\beta_1$ is unaffected by $\mathcal X_{div}$.
>
> **(iii) While $\beta_2$ is indeed influenced by $\mathcal X_{div}$, we can prove that $\beta_2$ is bounded by a very small constant. Therefore, in subsequent analyses, the effect of $\mathcal X_{div}$ on $\beta_2$ is negligible.** We have put the detailed discussion and proof in the official comment (1).
>
> ### **2) The parts of l.530 that are replaced do not seem to match the definition of  $\beta$'s.**
>
> **Upon careful examination, we believe the derivation is accurate.** However, we recognize that the presentation could be enhanced for clarity. We have provided a detailed derivation in the official comment (2).
>
> ---
> ## W3. Demonstrations of Theorems 2 and 3 seem to rely on one argument, which is: "Since $\mathcal X_{aux}\subset\mathcal X_{ood}$ , then $\mathcal H^*_{ood}\subset \mathcal H^*_{aux}$" I am concerned with the validity of this assumption.
>
> Thank you for your valuable feedback and we greatly appreciate your attention to detail. We illustrate the validity of this assumption as follow:
>
> **(i) The assumption is reasonable in the over-parameterized setting.**
>  Considering that we can decomposing the ideal error on $\mathcal X_{ood}$ into two components: one for $\mathcal X_{aux}$ and another for the remaining OOD data $\mathcal X_{ood} \setminus \mathcal X_{aux}$.
> Considering that the model is over-parameterized (line 126), there exists the ideal hypothesis minimizing both error simultaneously.
> In other word, $\mathcal X_{ood}$ is the intersection of the sets of optimal hypothesis for auxiliary and remaining data, thereby $\mathcal H^*_{ood}\subset\mathcal H^*_{aux}$. We have provided a detailed derivation in the official comment (3).
>
> **(ii) Discussion of the Counterexample.**
> The counterexample suggests that the model's optimal solution in $\mathcal X_{ood}$ may not be optimal in $\mathcal X_{aux}$ because it balances performance in $\mathcal X_{ood} \setminus \mathcal X_{aux}$. This assumes there is no hypothesis $h \in \mathcal H$ that achieves optimal performance in both $\mathcal X_{aux}$ and $\mathcal X_{ood} \setminus \mathcal X_{aux}$. However, in over-parameterized setting, there exists the ideal hypothesis that is optimal in both $\mathcal X_{aux}$ and $\mathcal X_{ood} \setminus \mathcal X_{aux}$. Therefore, **the counterexample does not hold in our setting**.
>
> ---
> ## 4. line 111 perhaps you meant $\mathcal X_{id}$ instead of $\mathcal Y_{id}$?
> ## 5. line 122 $\epsilon$ is called a probability, whereas it is an expectation.
> ## 6. Some typos
>
> Thank you for your valuable suggestions, we have addressed your concerns as follows:
>
> (i) We would like to clarify that $\mathcal X_{ood}$ is outside the support of $\mathcal Y_{id}$. In other words, $\mathcal X_{id}$ is within the support of $\mathcal Y_{id}$.
>
> (ii) We acknowledge the error on line 122 and have corrected it.
>
> (iii) We have thoroughly reviewed and revised all representation issues throughout the paper.
>
> References:
> >[1] The Loss Surface of Deep and Wide Neural Networks.
> >
> >[2] On the Quality of the Initial Basin in Overspecified Neural Networks.
> >
> >[3] Agree To Disagree: Diversity Through Dis-Agreement For Better Transferability.

---

> ### Author Response · Authors · 2024-08-07
> **(1) The proof about W2. Proof of the negligible effect of $\mathcal X_{div}$ on $\beta_2$**
>
> ## Proof of the negligible effect of $\mathcal X_{div}$ on $\beta_2$.
> Specifically, $\beta_1$ and $\beta_2$ represent the error of the ideal hypothesis on the unknown test-time data distribution $P_{\mathcal{X}}$ and the training data distribution $P_{\widetilde{\mathcal{X}}}$, respectively. We can derive that $\beta_2 \leq \beta_1$. Moreover, given that our model is over-parameterized, we have a sufficiently large hypothesis space to include a near-ideal hypothesis such that $\beta_1$ is sufficiently small, i.e., $\beta_1 \rightarrow 0$. As a result, we can conclude that $\beta_2 \rightarrow 0$. We have put the detailed proof in the official comment.
>
> **The detailed proof as follows:**
>
> Considering that $\beta_1$ and $\beta_2$ represent the error of the ideal hypothesis on the unknown test-time data distribution $P_{\mathcal{X}}$ and the training data distribution $P_{\widetilde{\mathcal{X}}}$, respectively, we have:
>
> $$
> \beta_2 = \underset{h \in \mathcal H}{\min} \epsilon_{x \sim \mathcal P_{\widetilde{\mathcal X}}}(h, f) = \min\limits_{h \in \mathcal H} \int_{\widetilde{\mathcal X}} |h(x) - f(x)|dx \\
> $$
>
> $$
> \beta_1 = \underset{h \in \mathcal H}{\min} \epsilon_{x \sim \mathcal P_{\mathcal X}}(h, f)= \min\limits_{h \in \mathcal H} \int_{\mathcal X} |h(x) - f(x)|dx
> = \min\limits_{h \in \mathcal H} \left( \int_{\widetilde{\mathcal X}} |h(x) - f(x)|dx + \int_{\mathcal X \setminus \widetilde{\mathcal X}} |h(x) - f(x)|dx \right)
> \ge \min\limits_{h \in \mathcal H} \int_{\widetilde{\mathcal X}} |h(x) - f(x)|dx + \min\limits_{h \in \mathcal H} \int_{\mathcal X \setminus \widetilde{\mathcal X}} |h(x) - f(x)|dx \ge \beta_2
> $$
>
> In our setting, the model is over-parameterized, meaning we have a sufficiently large hypothesis space to include a near-ideal hypothesis such that $\beta_1$ is sufficiently small. Therefore, we can denote $\beta_1 \rightarrow 0$. Given that $\beta_2 \leq \beta_1$, $\beta_2 \rightarrow 0$. Thus, $\beta_1$ and $\beta_2$ are both negligible, and we use a small value $\beta$ to unify them in Theorem 1.

---

> ### Author Response · Authors · 2024-08-07
> **(2) The proof about W2. The detailed proof from line 530 to the definition of $\beta$.**
>
> ## The detailed proof from line 530 to the definition of $\beta$.
>
> Let's first review line 530 on the demonstration of theorem 1:
> $$
> GError(h)\leq\epsilon_{x\sim\mathcal P_{\widetilde{\mathcal X}}}(h,f)+\textcolor{blue}{\epsilon_{x\sim\mathcal P_{\widetilde{\mathcal X}}}(h^*_{ood},f)}+\textcolor{blue}{\epsilon_{x\sim\mathcal P_{\mathcal X}}(h^*_{ood},f)} \\
> +\int |\phi_{\mathcal X}(x) -\phi_{\widetilde{\mathcal X}}(x) |\left| h(x)-h^*_{aux}(x) \right| \,dx
> +\epsilon_{x\sim\mathcal P_{{\mathcal X}}}(h^*_{aux},h^*_{ood}) \\
> +\textcolor{blue}{\epsilon_{x\sim\mathcal P_{\widetilde{\mathcal X}}}(h^*_{aux},f)}+\textcolor{blue}{\epsilon_{x\sim\mathcal P_{\widetilde{\mathcal X}}}(h^*_{ood},f)}
> $$
>
> We denote $\beta_1=\underset{h\in\mathcal H}{\min}\epsilon_{x\sim\mathcal P_{\mathcal X}}(h,f)$, $\beta_2=\underset{h\in\mathcal H}{\min}\epsilon_{x\sim\mathcal P_{\widetilde{\mathcal X}}}(h,f)$ as the error of $h^*_{ood}$ and $h^*_{aux}$ on $\mathcal P_{\mathcal X}$ and ${\mathcal P}_{\widetilde{\mathcal X}}$, respectively.
>
> In other word, for any $h\in \mathcal H^*_{ood}$, $\epsilon_{x\sim\mathcal P_{\mathcal X}}(h,f)=\beta_1$. For any $h\in \mathcal H^*_{aux}$, $\epsilon_{x\sim\mathcal P_{\widetilde{\mathcal X} }}(h,f)=\beta_2$.
>
> As a result, $\textcolor{blue}{\epsilon_{x\sim\mathcal P_{\mathcal X}}(h^*_{ood},f)=\beta_1}$, $\textcolor{blue}{\epsilon_{x\sim\mathcal P_{\widetilde{\mathcal X}}}(h^*_{aux},f)=\beta_2}$.
>
> Considering that $\mathcal H^*_{ood}\subset \mathcal H^*_{aux}$ (mentioned in line 126), for any $h\in \mathcal H^*_{ood}$, $h\in \mathcal H^*_{aux}$ is holds.
> As a result, $\textcolor{blue}{\epsilon_{x\sim\mathcal P_{\widetilde{\mathcal X}}}(h^*_{ood},f)=\beta_2}$. We have:
> $$
> GError(h)\leq \epsilon_{x\sim\mathcal P_{\widetilde{\mathcal X}}}(h,f)+\textcolor{blue}{\beta_2}+\textcolor{blue}{\beta_1}+\int |\phi_{\mathcal X}(x) -\phi_{\widetilde{\mathcal X}}(x) |\left| h(x)-h^*_{aux}(x)\right|dx +\epsilon_{x\sim\mathcal P_{\mathcal X}}(h^*_{aux},h^*_{ood})+\textcolor{blue}{\beta_2}+\textcolor{blue}{\beta_2}
> $$
> We denote $1/4*\beta=\max\\{ \beta_1,\beta_2\\}$, so:
> $$
> GError(h)\leq\epsilon_{x\sim\mathcal P_{\widetilde{\mathcal X}}}(h,f)+\int|\phi_{\mathcal X}(x)-\phi_{\widetilde{\mathcal X}}(x)||h(x)-h^*_{aux}(x)|dx+\epsilon_{x\sim\mathcal P_{\mathcal X}}(h^*_{aux},h^*_{ood})+\textcolor{blue}{\beta}
> $$

---

> ### Author Response · Authors · 2024-08-07
> **(3) The proof about W3. The detailed proof of the validity of the assumption.**
>
> ## The detailed proof of the validity of the assumption "Since $\mathcal X_{aux}\subset\mathcal X_{ood}$ , then $\mathcal H^*_{ood}\subset \mathcal H^*_{aux}$".
>
> For simplicity, we omit $\mathcal X_{id}$, assuming that the behavior is similar in this input space region for $h^*_{ood}$ and $h^*_{aux}$.
>
> We first express the expected error of hypotheses $h$ on the training data distribution $\mathcal P_{\widetilde{\mathcal X}}$ and the unknown test-time data distribution $\mathcal P_{\mathcal X}$ as follows:
>
> $$
> \begin{cases}
> \epsilon_{\mathcal P_{\widetilde{\mathcal X}}}(h, f) = \int_{\mathcal X_{aux}} |h(x) - f(x)|dx = \epsilon_1, \\\\
> \int_{\mathcal X_{ood} \setminus \mathcal X_{aux}} |h(x) - f(x)|dx = \epsilon_2, \\\\
> \epsilon_{\mathcal P_{\mathcal X}}(h, f) = \int_{\mathcal X_{ood}} |h(x) - f(x)|dx = \int_{\mathcal X_{aux}} |h(x) - f(x)|dx + \int_{\mathcal X_{ood} \setminus \mathcal X_{aux}} |h(x) - f(x)|dx = \epsilon_1 + \epsilon_2.
> \end{cases}
> $$
>
> From the above expressions, we obtain:
> $$
> \begin{cases}
> \mathcal{H}^*_{aux} = \\{ h : \arg \min\limits_h \epsilon_1 \\}, \\\\
> \mathcal{H}^*_{other} = \\{ h : \arg \min\limits_h \epsilon_2 \\}, \\\\
> \mathcal{H}^*_{ood} =\\{ h : \arg \min\limits_h (\epsilon_1 + \epsilon_2) \\}.
> \end{cases}
> $$
>
> The model is over-parameterized (line 126), which implies that our hypothesis space is large enough.
> Consequently, there exists the ideal hypothesis $h$ such that both $\epsilon_1$ and $\epsilon_2$ are minimized, i.e., $\mathcal{H}^*_{aux} \cap \mathcal{H}^*_{other} \neq \emptyset$. In this scenario, $\min_h (\epsilon_1 + \epsilon_2) = \min\limits_h \epsilon_1 + \min\limits_h \epsilon_2$. We denote $\mathcal{H}^*_{ood} = \mathcal{H}^*_{aux} \cap \mathcal{H}^*_{other}$, thus $\mathcal{H}^*_{ood} \subset \mathcal{H}^*_{aux}$.

---

> > ### Comment · Reviewer_1vuo · 2024-08-09
> > **Response to rebuttal**
> >
> > I would like to thank the authors for their detailed and structured responses! I am satisfied with responses to **W1**, **W2**-2 (please include the details in the final manuscript following these first two remarks), 5 and 6.
> >
> > ---
> >
> > I still have concerns about the rest. To start with the simpler:
> >
> > **4**: My concern is about the fact that the sentence suggests that the support of $\mathcal{Y_id}$ lives in the same space as $\mathcal{X_ood}$ and $\mathcal{X_id}$, which cannot be true. In addition, a support can be defined for a measure, a distribution (which I assumed was implicitly defined in $\mathcal{X_id}$ but after a double check, is not), or a function but not for a set. Instead, I would advise simply stating that $\mathcal{X_ood} = \mathcal{X} \backslash \mathcal{X_id}$
> >
> > **W2**-1:
> >
> > (i) How can you define $\beta$ as a constant if $\beta_1$ and $\beta_2$ are not and $\beta = 4 max(\beta_1, \beta_2)$?
> >
> > (ii -iii) I am afraid that the regime where $\beta_1 \rightarrow 0 $ and therefore $\beta_2 \rightarrow 0$ that you describe implies a "near-ideal hypothesis", where actually pretty much everything tends towards $0$ in the generalization bound 4. In other words, the problem I point out (that $\beta$'s are not constant) is only alleviated in a regime where Eq. 4 makes no longer sense in practice. What do you think about that?
> >
> > **W3**
> >
> > The overparametrized case happens when the number of parameters is larger than the number of training points. In that case, the model might perfectly fit the training points (interpolate), but nothing is guaranteed about the generalization error. In addition, you define your errors with continuous integrals, for which the model is never overparametrized because integrals can be defined as the limit of a sum of $n$ terms (1 for each data point) where $n\rightarrow \infty$ (Riemann definition).To obtain guarantees such as arbitrary minimization of the error, you should rely on universal approximation theorems, but they imply constraining assumptions for the underlying neural network (as a pointer see https://en.wikipedia.org/wiki/Universal_approximation_theorem and the references therein). These assumptions should be stated depending on the theorem you choose to use in your demonstration.

---

> > > ### Author Response · Authors · 2024-08-11
> > >
> > > Thank you sincerely for your detailed response and constructive feedback. Your insights have greatly contributed to our work, and we truly appreciate your support.
> > >
> > > ---
> > > We would like to further address your concerns as follows:
> > >
> > > ## W4. The support of $\mathcal Y_{id}$  lives in the same space as $\mathcal X_{ood}$ and $\mathcal X_{id}$, which cannot be true. Instead, I would advise simply stating that $\mathcal X_{ood}=\mathcal X  \setminus \mathcal X_{id}$.
> > >
> > > We sincerely appreciate your feedback, which has drawn our attention to the lack of preciseness in that definition. Your suggestion is indeed helpful. **we have decided to follow your advice and revise the definition of $\mathcal X_{ood}$ as follows:**
> > > $\mathcal X_{ood}=\mathcal X\setminus\mathcal X_{id}$ represent the input space of OOD data, where $\mathcal X$ represents the entire input space in the open-world setting.
> > >
> > > ## W2-1. (i) How can  you define $\beta$ as a constant if $\beta_1$ and $\beta_2$ are not and $\beta=4\max(\beta_1,\beta_2)$?
> > >
> > > Thank you for your comment, we appreciate the opportunity to address your concerns as follows:
> > >
> > > **(i) $\beta_1$ is a constant, given that $\beta_2 \leq \beta_1$, we have $\beta = 4\max(\beta_1, \beta_2) = 4\beta_1$, as a result, $\beta$ is a constant.**
> > >  Specifically, $\beta_1 = \underset{h \in \mathcal{H}}{\min} \epsilon_{x \sim \mathcal P_{\mathcal X}}(h, f)$, which depends on $\mathcal P_{\mathcal{X}}$ and $\mathcal{H}$, where
> > > $\mathcal{P}_{\mathcal{X}}$ represents the unknown test-time distribution in the open-world and does not change throughout our analysis. Similarly, $\mathcal{H}$ is a predefined hypothesis space that is fixed. Consequently, $\beta_1$ is a constant.
> > > As we derived in public comment (1), $\beta_2 \leq \beta_1$. Considering $\beta = 4\max(\beta_1, \beta_2) = 4\beta_1$, we can conclude that $\beta$ is a constant.
> > >
> > > **(ii) We recognize that our current presentation of $\beta = 4\max(\beta_1, \beta_2)$ may have led to some misunderstanding.** Our intention in introducing $\beta$ in Theorem 1 was to unify the small values $\beta_1$ and $\beta_2$. For coherence in our derivation, we use the definition of $\beta=4\max(\beta_1,\beta_2)$ directly. We appreciate your feedback and acknowledge that our current presentation could be clearer.
> > >
> > > **(iii) We have modified this part of the derivation in the revised version to enhance clarity.** Specifically, after proving $\beta_2 \leq \beta_1$ and clearly stating that $\beta_1$ is a constant, we have directly defined $\beta = 4\beta_1$. This modification should make our reasoning more transparent and easier to follow.
> > >
> > > ## W2-1(ii-iii) I am afraid that the regime where $\beta_1 \rightarrow 0$ and therefore that you describe implies a "near-ideal hypothesis", where actually pretty much everything tends towards 0 in the generalization bound 4. In other words, the problem I point out (that $\beta$'s are not constant) is only alleviated in a regime where Eq. 4 makes no longer sense in practice.
> > >
> > > We sincerely appreciate your feedback. We would like to address your concerns by first explaining the rationale behind our assumptions and then discussing the impact on Eq.4 in practice.
> > >
> > > **(i) We assume the existence of an ideal hypothesis $h$ within the hypothesis space $\mathcal{H}$ such that $\beta_1 \rightarrow 0$. According to universal approximation theorems, this condition can be met when the depth or width of deep neural networks satisfies certain conditions.** Specifically, under these conditions, the model becomes a universal approximator, implying the existence of $h^*_{ood} \in \mathcal{H}$ such that $h^*_{ood} \rightarrow f$, leading to $\beta_1 \rightarrow 0$.
> > >
> > > **(ii) In practical scenarios, Eq. 4 represents an upper bound on the generalization error of the learned hypothesis $h$. Moreover, when $\beta_1 \rightarrow 0$, each term in Eq. 4 retains its practical significance.**
> > > To illustrate this, let us revisit Eq. 4:
> > > $$
> > > \epsilon_{x\sim\mathcal P_{\mathcal X}}(h,f)
> > > \leq \underbrace{\hat \epsilon_{x \sim \mathcal P_{\widetilde {\mathcal X}}}(h,f)} _{\textbf{empirical error}}+\underbrace{\epsilon(h,h ^* _{aux})} _{\textbf{reducible error}}+\underbrace {\underset { h \in \mathcal H ^* _{aux}}{\sup }\epsilon _{x \sim \mathcal P _{\mathcal X}}(h,h ^* _{ood})} _{\textbf{distribution shift error}}+\underbrace {\mathcal R _m(\mathcal H)} _{\textbf{complexity}}+\sqrt{\frac{\ln(\frac{1}{\delta})}{2M}}+\beta
> > > $$
> > >
> > > where the empirical error term is minimized through optimization, the reducible error quantifies how closely $h$ approximates $h^*_{aux}$ and the distribution shift error captures the discrepancy between training and test data distributions. These components contribute significantly to the error upper bound. As $\beta_1 \rightarrow 0$, only the $\beta$ term (related to the ideal error) approaches zero, while the other terms remain relevant and unaffected.

---

> > > > ### Author Response · Authors · 2024-08-11
> > > >
> > > > ## W3. The overparametrized case happens when the number of parameters is larger than the number of training points. In that case, the model might perfectly fit the training points, but nothing is guaranteed about the generalization error. In addition, you define your errors with continuous integrals, for which the model is never overparametrized because integrals can be defined as the limit of a sum of terms (1 for each data point) where (Riemann definition). To obtain guarantees such as arbitrary minimization of the error, you should rely on universal approximation theorems, but they imply constraining assumptions for the underlying neural network. These assumptions should be stated depending on the theorem you choose to use in your demonstration.
> > > >
> > > > We sincerely appreciate your detailed response, thorough explanations, and constructive suggestions. Your input has significantly enhanced the theoretical rigor of our paper.
> > > >
> > > > **(i) We have acknowledged our misunderstanding regarding the overparameterized case.** To obtain guarantees such as the arbitrary minimization of error, we should indeed rely on universal approximation theorems.
> > > >
> > > > **(ii) We are now making specific assumptions about the underlying neural network to strengthen our proof.** Specifically, based on the paper [1], we assume our model is a fully-connected ReLU network with a width of (m+4), which can approximate any Lebesgue-integrable function $f$ from $\mathbb{R}^m$ to $\mathbb{R}$ with arbitrary accuracy with respect to $L^1$ distance. In this case, there exists an ideal hypothesis $h$ that minimizes both $\epsilon_1$ and $\epsilon_2$ simultaneously, i.e., $\mathcal{H}^*_{aux} \cap \mathcal{H}^*_{other} \neq \emptyset $, thus $\mathcal{H}^*_{ood} \subset \mathcal{H}^*_{aux}$.
> > > >
> > > > > [1] Universal Approximation Theorem for Width-Bounded ReLU Networks.
> > > >
> > > > ---
> > > > Finally, we would like to thank you again for your thorough review of our paper, your detailed feedback, and your constructive suggestions. They have significantly improved the quality of our work.

---

> > > > > ### Comment · Reviewer_1vuo · 2024-08-11
> > > > > **Final response to rebutal**
> > > > >
> > > > > Thank you for your perseverance and your honesty. I have final minor remarks that I do not expect the authors to respond to, but simply to take into account.
> > > > >
> > > > > - **Suggestion about the definition of $\beta$**: if you no longer use $\beta_2$ and directly bound it with $\beta_1$ (which is a cleaver way of alleviating concerns about $\beta_2$ not being a constant) I would recommend directly defining $\beta = 4 \int...$ rather than $\beta=4\beta_1$ for better readability.
> > > > > - **(Important) Assumptions for Universal Approximation Theorem:** Make sure to clearly state the assumptions in the main paper in the Theorem, not only in the proof.
> > > > >
> > > > > The authors have constructively engaged in a fruitful scientific discussion during this rebuttal, which has improved the quality of their manuscript and allowed to fix its initial theoretical imprecisions. As a result, I am happy to raise my score to acceptance.

---

> > > > > > ### Author Response · Authors · 2024-08-12
> > > > > > **Thank you again.**
> > > > > >
> > > > > > Thank you again for your valuable input and for increasing your rating. We will incorporate your valuable suggestions in the revised version to further improve the quality of our paper.

---

### Official Review · Reviewer_hmUc · 2024-07-09

**Soundness:** 3
**Presentation:** 3
**Contribution:** 2
**Rating:** 6
**Confidence:** 4

**Summary:**

The paper proposed Diversity-induced Mixup for OOD detection (diverseMix), which enhances the diversity of auxiliary outlier set for training in an efficient way.

**Strengths:**

1. The paper is written well and is easy to understand.

2. The studied problem is very important.

3. The results seem to outperform state-of-the-art.

**Weaknesses:**

1. My biggest concern is that there are already some papers that theoretically analyze the effect of auxiliary outliers and proposed some complimentary algorithms based on mixup, such as [1], [2] and [3]. It might be useful to clarify the differences.

[1] Out-of-distribution Detection with Implicit Outlier Transformation
[2] Learning to augment distributions for out-of-distribution detection
[3] Diversified outlier exposure for out-of-distribution detection via informative extrapolation

**Questions:**

see above

---

> ### Author Rebuttal · Authors · 2024-08-07
>
> Thank you for your reviews. We are encouraged that you appreciate our studied problem and state-of-the-art result. We address your concerns as follows.
>
> ## W1. My biggest concern is that there are already some papers that theoretically analyze the effect of auxiliary outliers and proposed some complimentary algorithms based on mixup, such as [1], [2] and [3]. It might be useful to clarify the differences.
>
> Thanks for mentioning [1] [2] [3]. We first review these related works and then clarify the differences from the perspectives of motivations, techniques, and theory. Additionally, we have incorporated these related works into the manuscript.
>
> **(i) Reviews of related works.**
>
> DOE [1] proposes a novel and effective approach for improving OOD detection performance by implicitly synthesizing virtual outliers via model perturbation.
>
> DAL [2] introduces a novel and effective framework for learning from the worst cases in the Wasserstein ball to enhance OOD detection performance.
>
> DivOE [3] is an innovative and effective method for enhancing OOD detection performance by using informative extrapolation to generate new and informative outliers.
>
> **(ii) Differences in motivations.** DiverseMix has a different motivation from DOE, DAL and DivOE. Our DiverseMix is proposed for enhancing the diversity of auxiliary outliers set  through semantic-level interpolation to enhance OOD detection performance. In comparison, DOE focuses on improving the generalization of the original outlier exposure by exploring model-level perturbation. DAL focus on crafting an OOD distribution set in a Wasserstein ball centered on the auxiliary OOD distribution to alleviating the distribution discrepancy between auxiliary outliers and unseen OOD data. DivOE focuses on extrapolating auxiliary outliers to generate new informative outliers for enhance OOD detection performance.
>
> **(iii) Differences in technique.** Our method, DiverseMix, has a different algorithmic design compared to others. Specifically, DiverseMix adaptively generates interpolation strategies based on outliers to create new outliers. In contrast, DOE, DAL, and DivOE primarily rely on adding perturbations. Specifically, DOE, DAL, and DivOE apply perturbations at the model level, feature level, and sample level, respectively, to mitigate the OOD distribution discrepancy issue.
>
> **(iv) Differences in theory.** We prove that a more diverse set of auxiliary outliers could improve the detection capacity from the generalization perspective, and this theoretical insight inspired our method DiverseMix. We also provide insightful theoretical analysis verifying the superiority of DiverseMix. In comparison, DOE revealing that model perturbation leads to data transformation  to enhance the generalization of OOD detector. DAL finds that the distribution discrepancy between the auxiliary and the real OOD data affecting the OOD detection performance. DivOE is based on the perspective of sample complexity to demonstrate its effectiveness.
>
> References:
> >[1] Out-of-distribution Detection with Implicit Outlier Transformation.
> >
> >[2] Learning to augment distributions for out-of-distribution detection.
> >
> >[3] Diversified outlier exposure for out-of-distribution detection via informative extrapolation.

---

> > ### Comment · Reviewer_hmUc · 2024-08-11
> > **Thank you!**
> >
> > Thanks for the clear response on my concerns and questions. All the things has been resolved, so I increase my score to 6. Thanks!

---

> > > ### Author Response · Authors · 2024-08-12
> > > **Thank you again.**
> > >
> > > Thank you so much for the valuable comments and increasing your rating. Thanks！

---

### Official Review · Reviewer_E172 · 2024-07-12

**Soundness:** 4
**Presentation:** 3
**Contribution:** 3
**Rating:** 6
**Confidence:** 4

**Summary:**

This study aims to explore the reasons behind the effectiveness of out-of-distribution (OOD) regularization methods by linking the auxiliary OOD dataset to generalizability. The researchers show that the variety within the auxiliary OOD datasets significantly influences the performance of OOD detectors. Moreover, they introduce a straightforward approach named diverseMix which is designed to enhance the diversity of the auxiliary dataset used for OOD regularization.

**Strengths:**

Strengths:
- The paper is well-composed and presents an extensive array of experiments across multiple OOD detection benchmarks.
- This study offers important insights into the significance of auxiliary datasets in OOD regularization, addressing a frequently neglected aspect of OOD regularization techniques.
- The authors provide a robust theoretical foundation for diverseMix. Additionally, they show strong empirical evaluations which further highlight the effectiveness of diverseMix across a range of OOD experiments.

**Weaknesses:**

Weakness:
- The reviewer has some concerns regarding the empirical evaluations of diverseMix. In particular, the choice of how the ImageNet-1k is split into ImageNet-200 as ID while the remaining data is leveraged as OOD seems arbitrary.

**Questions:**

The primary question of the reviewer is why not leverage the entire ImageNet-1k dataset for ID whilst leveraging other unlabeled datasets for the auxiliary data.

**Limitations:**

The authors have adequately addressed any potential negative societal impacts of this work, and the reviewer does not anticipate any such negative impacts.

---

> ### Author Rebuttal · Authors · 2024-08-07
>
> We sincerely thank the reviewer for your valuable comments and appreciate your recognition of the effective method as well as sufficient theoretical guarantees. We provide detailed responses to the constructive comments.
>
> ## W1. The reviewer has some concerns regarding the empirical evaluations  of diverseMix. In particular, the choice of how the ImageNet-1k is split into ImageNet-200 as ID while the remaining data is leveraged as OOD  seems arbitrary.
>
> Thank you for raising this important question. We sincerely appreciate your attention to the details of our experiments and pleasure to provide further clarification.
>
> (i) **Our splitting strategy was not arbitrary but carefully considered.** We randomly selected 200 classes from ImageNet-1K as the ID categories, while the remaining 800 classes were used as OOD categories. This setting ensures the validity of our experiments.
>
> (ii) **This experiment setting is consistency with prior work.** We followed the benchmark [1] to set ImageNet-200 as the ID dataset for our experiments.
>
> ## Q1. The primary question of the reviewer is why not leverage the entire  ImageNet-1k dataset for ID whilst leveraging other unlabeled datasets  for the auxiliary data?
>
> Thanks for the constructive suggestions. We are grateful for this insight and have conducted the additional experiments as suggested. Specifically, we used the entire ImageNet-1K dataset as the ID dataset and employed the SSB-hard dataset as auxiliary outliers. The experimental results are shown below, with columns representing different OOD datasets. The values in the table are presented in the format (FPR $\downarrow$ / AUROC $\uparrow$). **From the experimental results, it is evident that our method remains effective even when ImageNet-1K is used as the ID dataset.**
>
> | Method | dtd             | iNaturalist     | ninco           | average         |
> | :----: | :-------------: | :-------------: | :-------------: | :-------------: |
> | OE     | 73.90/76.82     | 49.33/89.53     | 76.03/80.52     | 66.42/82.29     |
> | Energy | 69.80/82.56     | 74.40/85.58     | 81.66/77.32     | 75.29/81.82     |
> | Mixoe | 69.48/78.07 | 46.61/89.72 | 74.17/80.79 | 63.42/82.86 |
> | Ours   | **68.17/78.69**   | **42.71/90.98**   | **73.29/81.27**   | **61.39/83.65**|
>
> References:
> > [1] OpenOOD v1.5: Enhanced Benchmark for Out-of-Distribution Detection.

---

### Comment · Area_Chair_A7QW · 2024-08-11
**Dear reviewers, please read and respond to authors' rebuttal (if you haven't done so)**

Dear reviewers, please read and respond to authors' rebuttal (if you haven't done so). Thanks.

Your AC

---

### Decision · Program_Chairs · 2024-09-25

**Decision:**

Accept (poster)

**Comment:**

This submission received three ratings (6, 6 and 7), averaging 6.33, which is above the acceptance. After rebuttal, all reviewers' concerns have been well addressed, especially about the experiments, difference from previous works and the clarity of the theoretical notations and proof. After carefully checking the response of all reviewers, I suggest the acceptance. Hope the authors carefully follow the reviewers' advice to improve the submission finally.